# Citrate serves as a signal molecule to modulate carbon metabolism and iron homeostasis in *Staphylococcus aureus*

**Feifei Chen**[1,2,3,4�]*, **Qingmin Zhao**[1,3,4�], **Ziqiong Yang**[1,3,4], **Rongrong Chen**[1,3,4], **Huiwen Pan**[1,3,4], **Yanhui Wang**[1,3,4], **Huan Liu**[1,3,4], **Qiao Cao**[1], **Jianhua Gan**[5], **Xia Liu**[3,6], **Naixia Zhang**[3,4], **Cai-Guang Yang**[1,3,4], **Haihua Liang**[2,7]*, **Lefu Lan**[1,2,8]*

1 School of Pharmaceutical Science and Technology, Hangzhou Institute for Advanced Study, University of Chinese Academy of Sciences, Hangzhou, China, 2 College of Life Science, Northwest University, Xi'an, China, 3 Shanghai Institute of Materia Medica, Chinese Academy of Sciences, Shanghai, China, 4 University of Chinese Academy of Sciences, Beijing, China, 5 State Key Laboratory of Genetic Engineering, Shanghai Public Health Clinical Center, Collaborative Innovation Center of Genetics and Development, School of Life Sciences, Fudan University, Shanghai, China, 6 Department of Diving and Hyperbaric Medicine, Navy Medical Center, Naval Medical University, Shanghai, China, 7 School of Medicine, Southern University of Science and Technology, Shenzhen, China, 8 Anhui Province Key Laboratory of Infectious Diseases, The First Affiliated Hospital of Anhui Medical University, Hefei, China

l These authors contributed equally to this work.
* feifeichen@simm.ac.cn (FC); lianghh@sustech.edu.cn (HL); llan@ucas.ac.cn (LL)

**Data Availability Statement:** *S. aureus* Newman genome can be found at National Center for Biotechnology Information with the accession number of NC_009641.1 (https://www.ncbi.nlm.

## Abstract

Pathogenic bacteria's metabolic adaptation for survival and proliferation within hosts is a crucial aspect of bacterial pathogenesis. Here, we demonstrate that citrate, the first intermediate of the tricarboxylic acid (TCA) cycle, plays a key role as a regulator of gene expression in *Staphylococcus aureus*. We show that citrate activates the transcriptional regulator CcpE and thus modulates the expression of numerous genes involved in key cellular pathways such as central carbon metabolism, iron uptake and the synthesis and export of virulence factors. Citrate can also suppress the transcriptional regulatory activity of ferric uptake regulator. Moreover, we determined that accumulated intracellular citrate, partly through the activation of CcpE, decreases the pathogenic potential of *S. aureus* in animal infection models. Therefore, citrate plays a pivotal role in coordinating carbon metabolism, iron homeostasis, and bacterial pathogenicity at the transcriptional level in *S. aureus*, going beyond its established role as a TCA cycle intermediate.

## Author summary

Citrate is the first intermediate of the TCA cycle, and its classic role is as a participant in metabolic reactions. In this study, we explored the non-metabolic role of citrate in *Staphylococcus aureus*. The results suggested that citrate can influence central carbon metabolism, iron homeostasis and the expression of virulence factors by modulating the regulatory activity of the catabolite control protein E (CcpE) and ferric uptake regulator (Fur). For instance, we revealed a new axis, named citrate-CcpE-*pycA*, was responsible for

nih.gov/nuccore/NC_009641.1/). Metabolomic data has been submitted to the database of MetaboLights (https://www.ebi.ac.uk/metabolights/) with the accession numbers MTBLS4276. RNA-seq data has been submitted to the NCBI Sequence Read Archive (SRA, https://ncbi.nlm.nih.gov/sra/) under the BioProject accession number PRJNA797550, with the BioSample accession numbers SAMN25010747 to SAMN25010758. IDAP-Seq data files were deposited to the NCBI Sequence Read Archive (SRA, https://ncbi.nlm.nih.gov/sra/) under the BioProject accession number PRJNA796052, with the BioSample accession numbers SAMN24812581 to SAMN24812584. Other data are included in the article and/or the supplementary information.

**Funding:** This work was supported by grants from National Key Research and Development Program of China (grant no. 2023YFD1800100 to LL), Hangzhou Institute for Advanced Study, UCAS (grant no. 2023HIAS-V006 to LL), National Natural Science Foundation of China (NSFC) (grant nos. 32270184 to LL and 31700124 to FC). The funders had no role in study design, data collection and analysis, decision to publish, or preparation of the manuscript.

**Competing interests:** The authors have declared that no competing interests exist.

controlling the endogenous biosynthesis of aspartate, which likely resulted in changes in metabolic adaptation in the host and influenced the pathogenesis of this important human pathogen. Thus, citrate serves as a key signaling molecule for the modulation of gene expression in *S. aureus*.

## Introduction

*Staphylococcus aureus* is one of the most important bacterial pathogens and has become an emerging concern of global public health [1]. This bacterium can colonize nearly all organs of the human body and cause a wide spectrum of infections, ranging from minor skin infections to life-threatening diseases such as bacteremia, endocarditis, pneumonia, toxic shock syndrome and sepsis [2]. However, currently, some *S. aureus* infections are difficult to treat because of the emergence of antibiotic-resistant strains such as methicillin-resistant *S. aureus* (MRSA), vancomycin-intermediate *S. aureus* (VISA), and vancomycin-resistant *S. aureus* (VRSA) [1,3].

The ability of *S. aureus* to cause a wide spectrum of infections has been linked to its ability to produce a variety of virulence factors [4,5]. This includes a plethora of proteases, toxins, adhesins, and immune evasion factors [5]. Expression of these virulence factors is tightly coordinated by a large array of global regulators, including two-component systems and SarA family proteins [6,7]. Recently, there is increasing evidence that metabolism and bacterial pathogenicity are intertwined [8] and it is believed that metabolic flexibility is essential for the pathogenicity and virulence of *S. aureus* [9]. Indeed, genes involved in metabolic pathways, particularly those of central carbon metabolism (e.g., glycolysis, gluconeogenesis, the pentose phosphate pathway and TCA cycle), have been identified as important determinants of *S. aureus* survival during *in vivo* infections [10,11]. It is also becoming apparent that *S. aureus* has integrated its virulence gene regulation with the direct or indirect sensing of metabolites by transcriptional regulators (i.e., CcpE, CcpA, CodY, RpiRc, and PurR) [12–20]. To date, how metabolism affects the pathogenicity of *S. aureus* remains an active area of research, and a deep understanding of the underlying mechanism may provide new strategies to treat or prevent infections caused by this notorious pathogen [9].

We previously showed that *S. aureus* CcpE, a LysR-type transcriptional factor, senses and responds to citrate, an intermediate of the TCA cycle [12](Fig 1A). In addition to serving as a positive regulator of *citB*, the gene encoding the TCA cycle aconitase, CcpE also influences the expression of genes involved in toxin production and capsule biosynthesis and acts as a negative regulator of *S. aureus* virulence [12]. Based on these previous findings, we aimed to investigate the role of citrate in *S. aureus* cellular functions and the underlying mechanisms in this study. We found that citrate accumulation leads to the activation of CcpE and inhibition of ferric uptake regulator (Fur), resulting in extensive changes in the metabolome and transcriptome of *S. aureus* and reduced pathogenicity. Our results highlight that citrate functions as a signaling metabolite regulating various cellular processes at a transcriptional level.

## Results

### Citrate can influence the metabolome of *S. aureus* independent of the TCA cycle

To evaluate the effect of citrate on metabolite production in *S. aureus*, we utilized ultra-high-performance liquid chromatography coupled with hybrid triple quadrupole time-of-flight

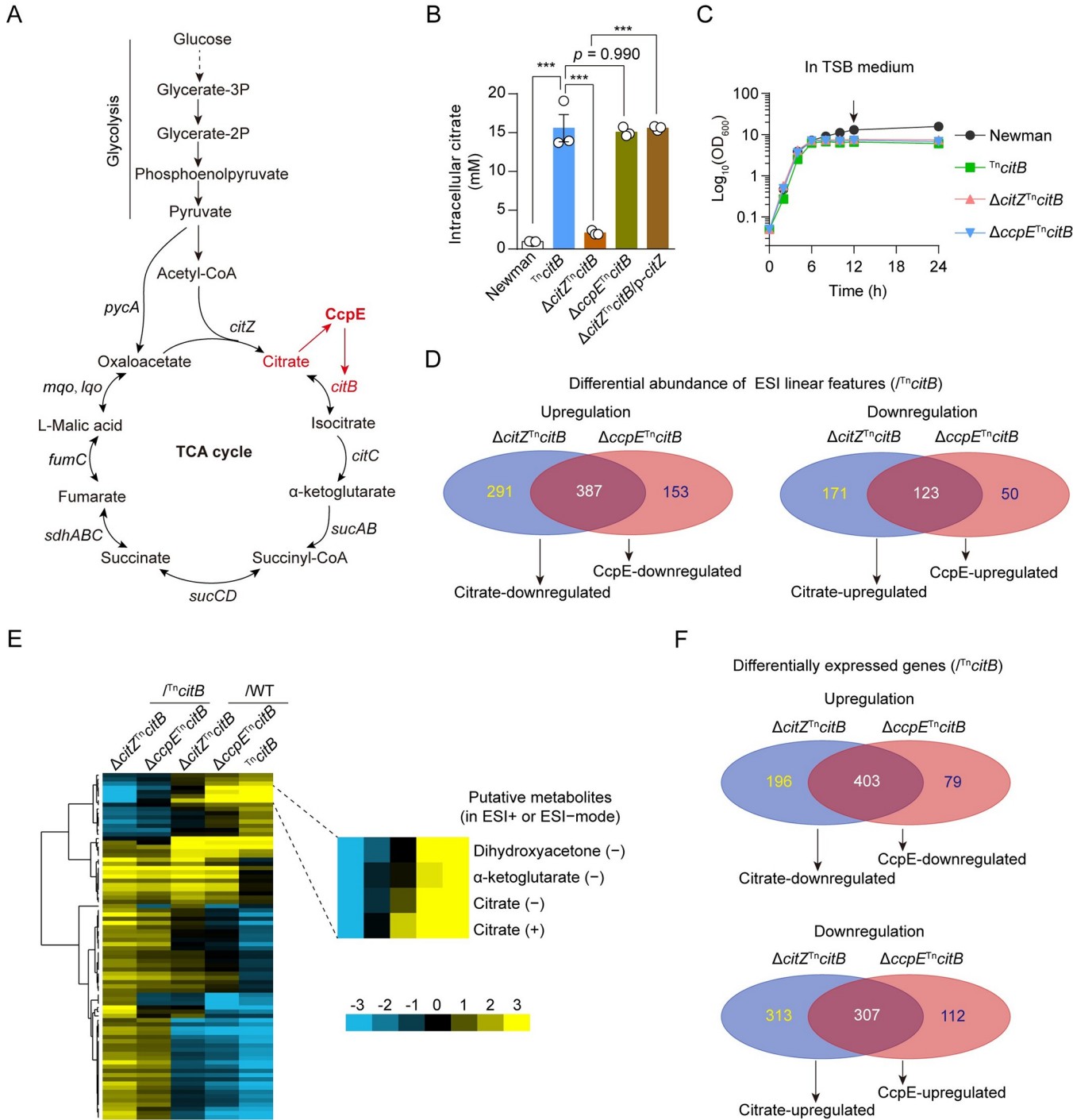

**Fig 1. Effect of *citZ* or *ccpE* deletion on the metabolome and transcriptome of the $^{Tn}$*citB* mutant.** (A) A brief scheme shows the glycolysis pathway and the TCA cycle of *S. aureus*. Citrate is an activator of CcpE, which positively regulates the expression of *citB* (shown in red). (B) The intracellular citrate levels of *S. aureus* Newman and its isogenic mutants cultured in TSB medium for 6 h. Newman, $^{Tn}$*citB*, $\Delta$*citZ*$^{Tn}$*citB*, and $\Delta$*ccpE*$^{Tn}$*citB* harbor an empty control vector, respectively. $\Delta$*citZ*$^{Tn}$*citB*/p-*citZ* represents the $\Delta$*citZ*$^{Tn}$*citB* strain complemented with *citZ*. (C) Growth curve of Newman and its isogenic mutants cultured in TSB were measured via optical density at 600 nm. Arrow indicated the time at which cultures were sampled for metabolomics and RNA-seq analysis. (D) Venn diagram indicates the overlap between citrate- and *ccpE*-regulated metabolic features (in ESI$^+$ and/or ESI$^-$ modes). The terms "citrate-regulated" and "CcpE-regulated" refer to the alterations (VIP≥1 and *p*-value < 0.05) detected in the $\Delta$*citZ*$^{Tn}$*citB* strain sample and the $\Delta$*ccpE*$^{Tn}$*citB* strain sample, respectively, as compared to the $^{Tn}$*citB* strain sample. (E) A heatmap shows the relative abundance of 102 citrate-regulated metabolites. Values expressed as log$_2$fold-change. (F) Venn diagram indicates the overlap between citrate- and *ccpE*-regulated genes. The terms "citrate-regulated" and "CcpE-regulated" refer to the alterations (≥ 2-fold change and *p* < 0.05) detected in the $\Delta$*citZ*$^{Tn}$*citB* strain sample and the $\Delta$*ccpE*$^{Tn}$*citB* strain sample, respectively, as compared to the $^{Tn}$*citB* strain sample. In (B) and (C), data represents mean ± SD from *n* = 3 biological replicates. *** *p* < 0.001 by One-way ANOVA with Tukey test (B).

mass spectrometry (UHPLC-Q-TOF-MS)-based metabolome analysis. Specifically, we analyzed the wild-type (WT) Newman strain and its isogenic Tn*citB* mutant (with a transposon insertion in *citB*), taking advantage of the fact that the Tn*citB* mutant had an intracellular citrate concentration approximately 16 times higher than that of the WT *S. aureus* Newman (Fig 1B). Given that inactivating *citB* elevates intracellular citrate levels while also disrupting the TCA cycle [21], we incorporated a double knockout mutant of both *citZ* and *citB* (Δ*citZ*Tn*citB*) into our study. In this mutant, the TCA cycle is also impaired, but intracellular citrate levels are significantly lower (~ 2 mM) compared to those in the Tn*citB* mutant (Fig 1B). Furthermore, as citrate is a recognized activator of CcpE [12](Fig 1A), a double knockout mutant of both *ccpE* and *citB* (Δ*ccpE*Tn*citB*), was included in the study. This mutant, with comparable intracellular citrate levels to the Tn*citB* mutant (Fig 1B), was utilized to explore the involvement of CcpE in the regulatory role of citrate.

Unexpectedly, we observed that the intracellular citrate levels of the Δ*citZ*Tn*citB* mutant were similar to those of the WT Newman strains (Fig 1B). We hypothesized that SbnG, another citrate synthase in *S. aureus*, might contribute to the intracellular citrate pool in the Δ*citZ*Tn*citB* mutant. Deletion of *sbnG* did not significantly affect the intracellular citrate levels of the Δ*citZ*Tn*citB* mutant cultured in TSB medium (S1A Fig). However, when cultured in tryptic soy broth (TSB) containing 1mM iron chelator 2,2′-dipyridyl (DIP), the Δ*sbnG*Δ*citZ* Tn*citB* triple mutant showed an undetectable level of intracellular citrate (S1B Fig). In contrast, the complemented strain, Δ*sbnG*Δ*citZ* Tn*citB*/p-*sbnG*, exhibited comparable intracellular citrate levels to the Δ*citZ*Tn*citB* mutant (S1B Fig), suggesting that SbnG plays a crucial role in regulating the intracellular citrate levels of the Δ*citZ*Tn*citB* mutant under iron deprivation conditions. The exact source of intracellular citrate in the Δ*citZ*Tn*citB* mutant cultured in iron-replete TSB medium remains unknown, warranting further investigation to clarify this aspect.

After growth of *S. aureus* strains in TSB for 12 hours (Fig 1C), we identified a total of 13,314 metabolic features, comprising 6,592 features in ESI+ mode and 6,722 features in ESI-mode, in samples obtained from four testing *S. aureus* strains: WT Newman, Tn*citB*, Δ*citZ*Tn-*citB*, and Δ*ccpE*Tn*citB* (Sheets A to C of S1 Table). The orthogonal partial least squares-discriminant analysis (OPLS-DA) showed clustering of pooled quality control (QC) samples (S2A and S2B Fig), confirming the stability and reproducibility of the instrumental analysis. Analysis of the Δ*citZ*Tn*citB* strain samples revealed 972 differential metabolic features (in ESI$^+$ and/or ESI$^-$ modes) compared to Tn*citB* cells (VIP$\geq$1 and *p*-value $< 0.05$), with 678 features up-regulated and 294 features down-regulated (Fig 1D and Sheet A of S1 Table).

Among the 678 up-regulated features, 387 were found to be up-regulated upon the deletion of *ccpE* in the Tn*citB* mutant (Fig 1D). Meanwhile, of the 294 down-regulated features, 123 were also found to be down-regulated in the Tn*citB* mutant as a result of the deletion of *ccpE* (Fig 1D and Sheet A of S1 Table).

Using database alignment of the measured masses, we were able to identify a total of 102 metabolites that were regulated by citrate, which corresponded to 128 metabolic features (71 in ESI+ mode and 57 in ESI- mode) (Fig 1E and Sheets D to F of S1 Table). Citrate itself was present in both ESI+ and ESI- mode (Fig 1E and Sheets D to F of S1 Table), with a production pattern similar to bacteria grown in TSB medium for 6 hours (Fig 1B). Approximately 50% of the 30 citrate-upregulated metabolites were intermediates of central carbon metabolism (such as dihydroxyacetone, dihydroxyacetone phosphate, 2-Phosphoglycerate, 3-Phospho-D-glycerate, and alpha-ketoglutarate), nicotinamide-adenine dinucleotide phosphate (NADP), and nucleotides (such as ATP, UTP, CDP, UDP, GDP, UMP) (Sheets D to F of S1 Table). In contrast, over 50% of the 72 citrate-downregulated metabolites consisted of amino acids and their derivatives (e.g., dipeptides) (Sheets D to F of S1 Table). Collectively, these findings suggest

that the accumulated citrate impacts the metabolome of the $^{Tn}citB$ mutant, and this effect is primarily mediated by the activation of CcpE.

## Citrate can change *S. aureus* transcriptome independently of TCA

Through RNA-Seq experiments, we discovered that the deletion of *citZ* led to alterations in the expression levels of 1,219 genes in the $^{Tn}citB$ mutant ($\geq$ 2-fold change and $p < 0.05$) (S2 Table). Among these genes, 599 were up-regulated and 620 were down-regulated (Fig 1F). These genes accounted for around 41% of *S. aureus* Newman genes and were associated with various biological processes (S2 Table). Deletion of the *citZ* in $^{Tn}citB$ mutant resulted in significant down-regulation (>16 fold) of genes encoding virulence factors (i.e., *hla* and *scn*), phosphotransferase system (PTS) (i.e., *NWMN_RS01850-NWMN_RS01835* operon, *NWMN_RS14005*), *aa3*-type terminal oxidase (*qoxABCD* operon), and transcriptional regulators (i.e., *cstR*, *farR* and *sarR*) (S2C Fig and S2 Table). In contrast, transcription of *clpC* operon (i.e., *ctsR-mcsA-mcsB-clpC*) and those encoding acetoin reductase ButA and heat-inducible transcription repressor HrcA were dramatically increased (> 16-fold) (S2C Fig and S2 Table). These findings indicate that intracellular citrate accumulation contributes to significant changes in *S. aureus* gene expression.

Furthermore, through RNA-Seq experiments, we found that removing *ccpE* from the $^{Tn}citB$ mutant strain led to changes in the expression of 901 genes ($\geq$ 2-fold change and $p < 0.05$) (S2 Table). Intriguingly, consistent with the results of the metabolome analysis (Fig 1D), the majority of differentially expressed genes were concurrently regulated by citrate and CcpE (Fig 1F). Thus, the removal of *ccpE* from the $^{Tn}citB$ mutant has a comparable, yet not identical, impact on the transcriptome as the deletion of *citZ*, providing further support and evidence for the activation of CcpE by citrate.

We conducted a differential networking analysis using Joint-Pathway Analysis [22] to further our investigation. This integrated analysis of transcriptomics and metabolomics facilitated a comprehensive visualization of the interactions between differentially expressed genes and perturbed metabolites. By integrating the data, we identified 11 and 8 enriched pathways (with $p$-value $< 0.05$ and an impact value greater than 1) in the $\Delta citZ^{Tn}citB$ (S3A Fig) and $\Delta ccpE^{Tn-}citB$ (S3B Fig) mutants, respectively, as compared to the $^{Tn}citB$ mutant (S3 Fig). Notably, all of the enriched pathways identified in the $\Delta ccpE^{Tn}citB$ mutant were in complete concordance with those found in the $\Delta citZ^{Tn}citB$ (S3 Fig). These pathways include pyruvate metabolism, purine metabolism, alanine, aspartate and glutamate metabolism, arginine biosynthesis, glycine, serine and threonine metabolism, pyrimidine metabolism, glycolysis or gluconeogenesis, and pentose phosphate pathway (S3 Fig).

## Genome-wide identification of CcpE targets

As CcpE regulates the expression of numerous genes in *S. aureus* either directly or indirectly, we conducted *in vitro* DNA affinity purification and sequencing (IDAP-Seq) experiments [23] to investigate its direct targets. Using the peak-calling software MACS2, we identify 390 peaks (fold enrichment $\geq$ 2 and $-\log_{10}^{q\text{-value}} > 50$) from three independent replicates and found 107 peaks were reproducible in at least two independent experiments (Sheet B of S3 Table). Out of the 390 enriched peaks, 238 (61%) had peak summits located at intergenic regions (Sheet A of S3 Table), which constitute only 16.6% of the *S. aureus* Newman genome [24]. The substantial enrichment of CcpE-binding sites in these intergenic regions, where regulatory sites are commonly located, supports CcpE's role as a transcription regulator.

The enriched peaks for CcpE displayed a broad occupancy over the *S. aureus* Newman chromosome (Fig 2A and Sheet A of S3 Table). The promoter region of pyruvate carboxylase

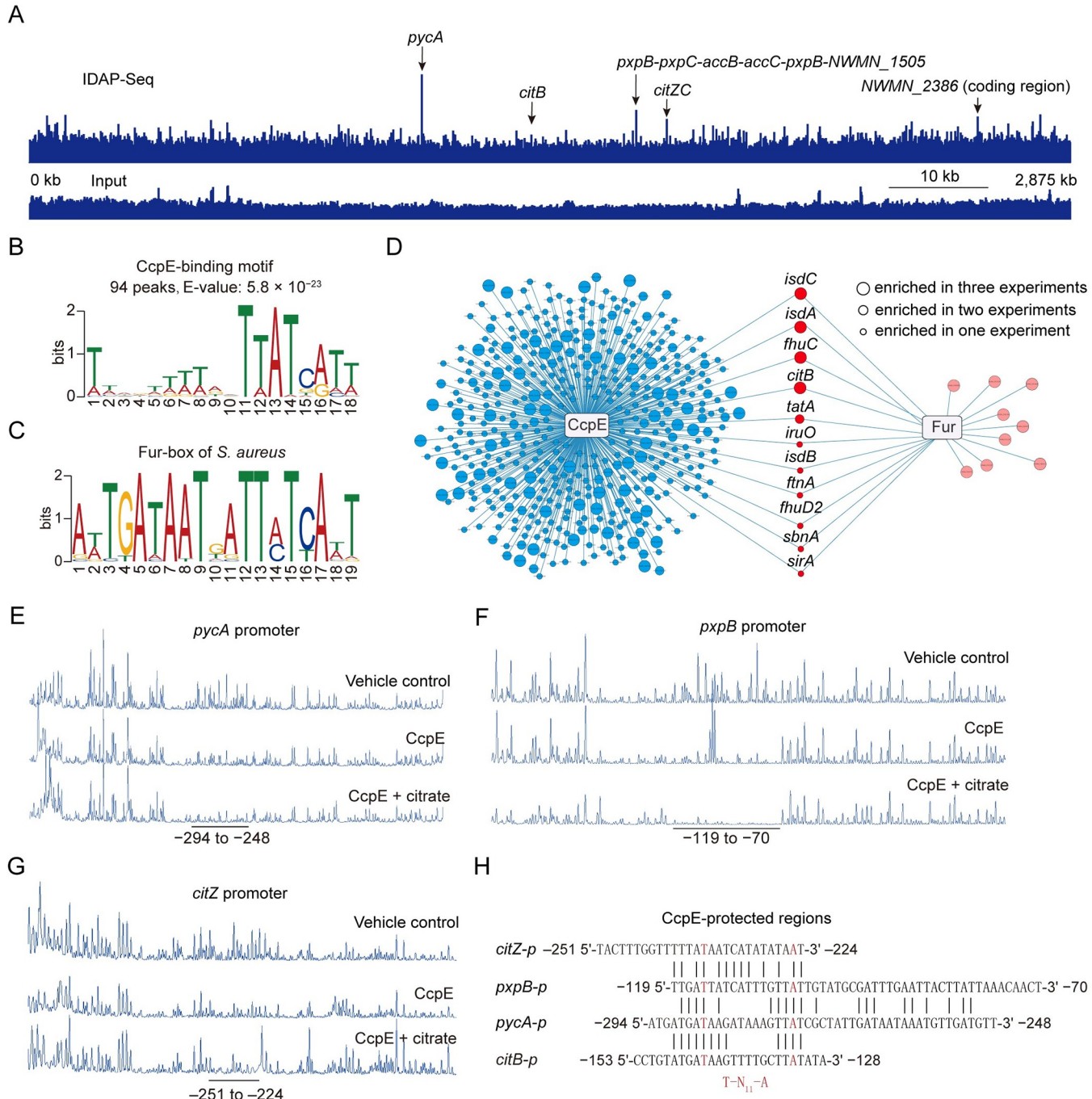

**Fig 2. Identification of CcpE targets.** (A) Representative image of the CcpE IDAP-seq data illustrated by Integrated Genome Viewer (IGV) software. The black arrows depicted the top five ChIP-Seq peaks identified in the promoters of *pycA*, *citB*, *pxpB* operon, and *citZC* operon, and the coding region of *NWMN_2386*. (B) CcpE-binding motif derived from a MEME analysis of a set of 81-bp DNA sequences centered at the summit from 107 reproducible IDAP-seq peaks. (C) Fur-binding motif derived from 23 Fur-recognized sequence in *S. aureus*. (D) Visualization of promoters bounded by CcpE and Fur. Blue circles denote promoters bounded by CcpE, red circles denote promoters co-targeted by CcpE and Fur, and pink circles denote promoters targeted by Fur. The lines illustrate the connection between transcription factors and their respective target promoters. (E to G) Electropherograms showed the protection pattern of *pycA*, *pxpB*, and *citZ* promoter DNA after digestion with DNase I following incubation without or with CcpE in the absence or presence of citrate. Regions protected (relative to the start codon) by CcpE are underlined. (H) Alignment of the CcpE-protected DNA sequences in the promoter of *pycA* (E), *pxpB* (F), *citZ* (G) and *citB*. The potential LTTR box (T-$N_{11}$-A, where N is any nucleotide) are highlighted in red.

gene (*pycA*) exhibited the highest average fold-enrichment (23.7-fold) among the enriched peaks (S4A Fig and Sheet A of S3 Table). The four next most enriched peaks (ranked by average fold-enrichment) were found in the promoter region of a six-gene operon, including 5-oxoprolinase genes (i.e., *pxpB*, *pxpC*, and *pxpA*) and acetyl-CoA carboxylase genes (i.e., *accB* and *accC*) (S4B Fig), the promoter region of *citZC* operon (comprising two TCA cycle genes, *citZ* and *citC*) (S4C Fig), the coding region of *NWMN_2386* (S4D Fig), and the promoter region of *citB* (S4E Fig). Notably, the *citB* promoter has been previously identified as a direct target of CcpE [12,25], supporting the effectiveness of our IDAP-seq procedure.

Upon annotating the 390 CcpE-enriched peaks based on their positions, it was revealed that 682 genes within 443 predicted transcription units (TUs) might potentially be regulated by CcpE (Sheet A of S3 Table). According to functional analyses, these potential CcpE-targeted genes are associated with a variety of biological processes, such as central carbon metabolism (e.g., *pycA*, *zwf*, *accB* and acc*C*, *citZC* operon, *citB*, *maeB*), iron uptake and homeostasis (e.g., *isdA*, *isd* operon, *isdI*, *fhuABG* operon, *tatAC* operon, *sufCDSUB* operon), stress response (e.g., *sodM*, *dsbA*, *copZ*) and the synthesis or export of virulence factors (e.g., *pmt* operon, *capABC* operon, *tagA*, *tagH*, *tarGBXD* operon, *tarIJLS* operon, *sasD*, *csa1A*, and *nuc*) (Sheet A of S3 Table).

After comparing the genes potentially targeted by CcpE, which were obtained from IDAP-Seq experiments, with RNA-seq results (i.e., Δ*ccpE*^Tn*citB*/^Tn*citB*), we discovered that CcpE regulates the expression of 218 genes (in 141 predicted transcription units) responsible for diverse cellular functions (Sheet A of S3 Table). Notably, CcpE binds to the promoters and modulates the expression of genes involved in numerous metabolic processes, such as amino sugar metabolism (e.g., *nanA*, *nanK*, *nanE*, *glmS*), gluconeogenesis (e.g., *fbp*), pentose phosphate pathway (e.g., *zwf*, *tal*), fatty acids synthesis (e.g., *accB*, *accC*), the TCA cycle (e.g. *citB*), amino acid metabolism (i.e., *pxpA*, *pxpB*, *pxpC*, *metL*), and the phosphoenolpyruvate (PEP)-pyruvate-oxaloacetate (PPO) node (e.g., *pycA*), an area located at the intersection of glycolysis and the TCA cycle (S5 Fig) [26]. However, the detailed mechanisms by which CcpE affects diverse metabolic processes remain to be defined.

Using the MEME suite [27], we identify an 18-bp A/T-enriched motif, TWWDWWWWRVTTATCATT (W = A or T, D = G or A or T, R = A or G, V = A or C or G), in a subset of CcpE-targeted DNAs (Fig 2B). Specifically, it was present in 94 out of the 107 reproducible peaks (Fig 2B and S4 Table). Interestingly, we observed a significant similarity between the *in silico* DNA motif recognized by CcpE and the ferric uptake regulator (Fur), with a statistically significant *p*-value of $7.42 \times 10^{-3}$, as calculated by Motif Comparison Tool with a Pearson correlation coefficient comparison (Fig 2C). In support of our findings, 11 out of 20 known Fur-regulated promoters [28,29] were directly bound by CcpE (Fig 2D and Sheet A of S3 Table). Taken together, these results provide insights into the regulatory mechanisms of CcpE and suggest the possibility of cross-talk between CcpE and Fur in the regulation of genes.

## CcpE inhibits the expression of *pycA*

As mentioned above, CcpE binds to promoter and modulates the expression of various metabolic gene, including *pycA* that encodes the pyruvate carboxylase (S4A and S5 Figs), which is a crucial phosphoenolpyruvate (PEP)-pyruvate-oxaloacetate node (PPO-node) enzyme [26] and a determinant factor of *S. aureus* pathogenicity [11,30]. To validate the binding of CcpE to the promoter of *pycA*, DNase I footprinting analyses were conducted. The results showed that CcpE indeed binds to the promoter of *pycA* (Fig 2E). Additionally, through comparison of the CcpE-protected region of the *pycA* promoter with those of *pxpB* (Fig 2F), *citZ* (Fig 2G), or *citB*

promoter [12], we have identified a potential T-$N_{11}$-A motif (Fig 2H) that are recognized by LysR family proteins [31]. Through 5'-rapid amplification of cDNA ends (5'-RACE) experiments, we determined that the transcriptional start site of *pycA* is located 275 bp upstream of its start codon, specifically at the A site of the potential T-$N_{11}$-A motif. When the T-$N_{11}$-A motif of the *pycA* promoter was mutated to G-$N_{11}$-G, the inhibitory impact of *ccpE* on the *pycA* promoter (Fig 3A and 3B) disappeared (Fig 3C and 3D). Disrupting the T-$N_{11}$-A motif led to a loss of upregulation in both wild-type and Δ*ccpE* cells (Fig 3C and 3D), indicating that the T-$N_{11}$-A motif is not only crucial for the regulation of *pycA* by CcpE but also plays a significant role in the promoter activity of *pycA*. Using *in vitro* transcription assays, we demonstrated that CcpE can suppress *pycA* transcription (Fig 3E) while activating transcription of *citB* (Fig 3F). Altogether, these results indicate that CcpE serves as a repressor for *pycA*.

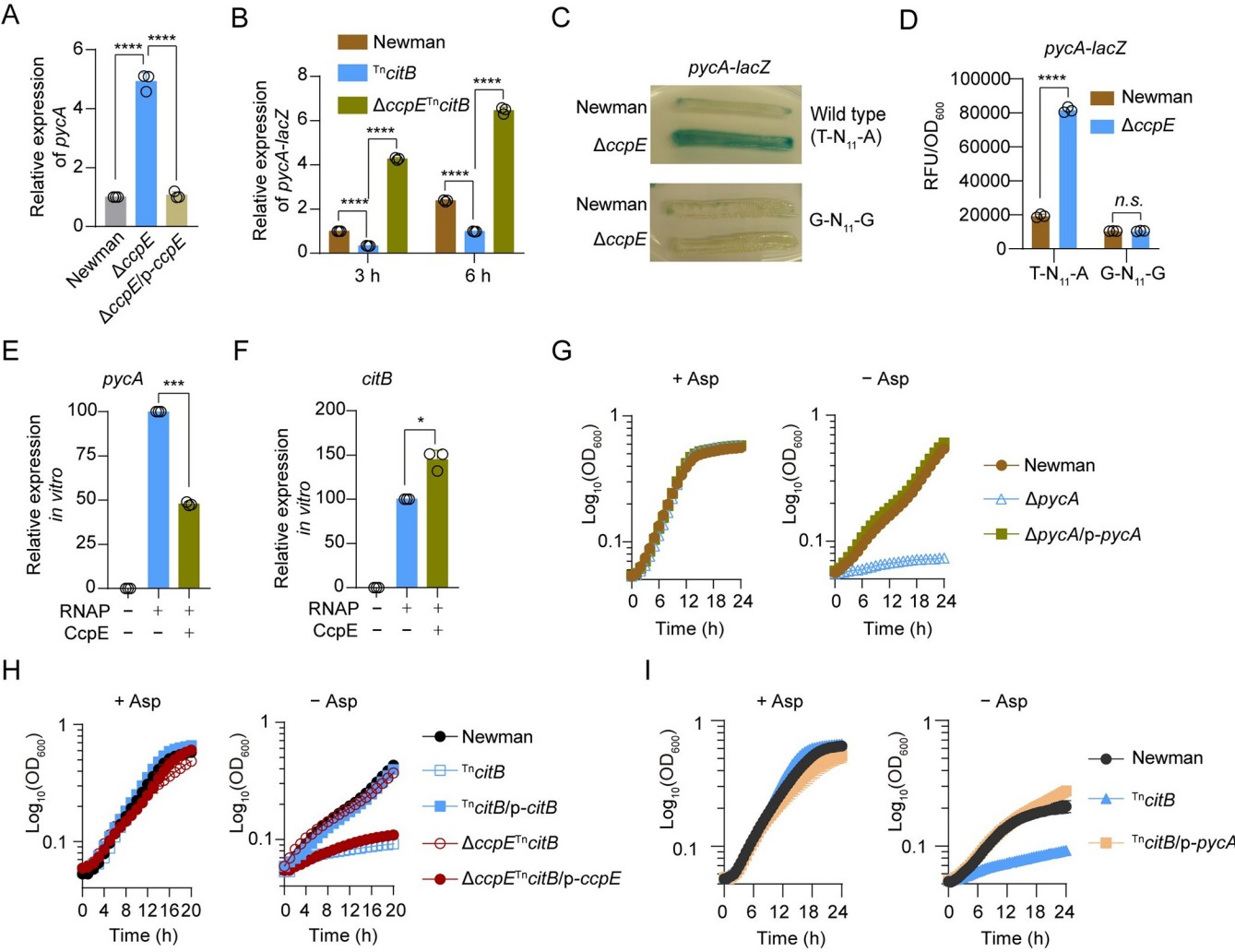

**Fig 3. CcpE inhibits the expression of *pycA*.** (A) Effect of *ccpE* deletion on the expression of *pycA* in *S. aureus* Newman cultured in TSB medium for 6 h. ****$p < 0.0001$, by one-way ANOVA with Dunnett test. (B) Relative *pycA-lacZ* activity in *S. aureus* strains grown in TSB. ****$p < 0.0001$, by two-way ANOVA with Tukey test. (C) Representative image of *pycA-lacZ* activity in *S. aureus* strains grown on TSA plate containing 160 μg/mL X-gal for 24 h. Strains carry either the wild-type or a mutated G-$N_{11}$-G box motif of the *pycA* promoter. (D) *pycA-lacZ* activity grown in TSB for 4.5 h. The relative fluorescence intensity (RFU) was normalized to the optical density of bacteria at 600 nm ($OD_{600}$). *n.s.*, $p>0.05$, ****$p < 0.0001$, by two-way ANOVA with Sidak test. (E and F) Effect of CcpE on the transcription of *pycA* (E) and *citB* (F) *in vitro*, *$p < 0.05$ and ***$p < 0.001$, by two-tailed one-sample *t*-test. (G to I) Bacterial growth curves in chemical defined medium (CDM) supplemented with (+ Asp) or without (- Asp) the addition of aspartate (2.4 g/L). Newman, Δ*pycA*, $^{Tn}$*citB* and Δ*ccpE*$^{Tn}$*citB* harbor an empty control plasmid, respectively. In (A), (B), (D), and (G to I), data represents mean ± SD from $n = 3$ biological replicates. In (E) and (F), data represents mean ± SD from $n = 3$ independent experiments.

In *S. aureus*, PycA supplies oxaloacetate, which serves as a precursor for aspartate, a central building block for many metabolic processes such as biosynthesis of other amino acids (S5 Fig)[32]. Deletion of *pycA* made *S. aureus* Newman almost unable to grow in chemically defined medium (CDM) without aspartate (Asp) (Fig 3G). To further confirm the role of CcpE in regulating the biological function of *pycA*, the ability of WT *S. aureus* Newman, $^{Tn}citB$, and $\Delta ccpE^{Tn}citB$ strains to grow in the CDM supplemented with or without Asp was tested. Our results indicated that both $^{Tn}citB$ and $\Delta ccpE^{Tn}citB$ strains grew similarly to the WT Newman strain in CDM that contains an abundance of amino acids, including Asp (Fig 3H). However, when Asp was excluded from the medium, the growth of $^{Tn}citB$, but not $\Delta ccpE^{Tn-}citB$, was notably lower than that of the WT *S. aureus* Newman strain (Fig 3H). Complementation of the $\Delta ccpE^{Tn}citB$ mutant with a plasmid-borne *ccpE* made it almost unable to grow in CDM without Asp (Fig 3H), suggesting that the activation of CcpE plays a crucial role in the aspartate auxotrophy observed in the $^{Tn}citB$ mutant. Moreover, the constitutive expression of *pycA* fully restored the growth of the $^{Tn}citB$ mutant cultured in CDM without Asp to a level comparable to that of the wild-type Newman strain (Fig 3I). Collectively, these results suggest that activation of CcpE, which lead to decreased expression of *pycA*, may be the cause of Asp auxotrophy observed in the $^{Tn}citB$ mutant.

## CcpE competes with Fur for binding to the intergenic region of *isdA* and *isdC*

CcpE binds to the promoters and enhances the expression of well-known iron-regulated genes (S6A–S6D Fig and Sheet A in S3 Table) including those (i.e., *isdA*, *isdC*, *isdD*, and *isdI*) encoding the iron-regulated surface determinant (Isd) system [33]. In *S. aureus*, *isdA* is transcribed in the opposite direction from *isdC*, which is the first gene in the *isdCDEF-srtB-isdG* operon (Fig 4A) [33]. The transcription of *isdA*, *isdCDEF-srtB-isdG*, and three other *isd* genes (*isdB*, *isdI*, and *isdH*) is known to be regulated by Fur through conserved DNA sites called Fur-boxes (Fig 4A) [33]. Using *in vitro* transcription experiments, we demonstrated that CcpE is a direct activator of *isdC* (Fig 4B). Using DNase I footprinting experiments, we showed that CcpE binds to the intergenic region of *isdA* and *isdC* and that the CcpE-protected regions overlap with those of Fur (Fig 4C). This finding is consistent with our previous outcomes indicating that the *in silico* DNA motif recognized by CcpE is comparable to that of Fur (Fig 2B and 2C). Furthermore, through EMSA experiments, we showed that CcpE could compete with Fur for binding to the intergenic region of *isdA* and *isdC* (Fig 4D), and vice versa (S7A Fig)

To verify the regulatory influence of CcpE on the Isd system in *S. aureus*, we conducted Western blot analyses on various strains, including the WT Newman strain and its derivatives (i.e., $^{Tn}citB$, $\Delta citZ^{Tn}citB$, $\Delta citZ^{Tn}citB$/p-*citZ*, $\Delta ccpE^{Tn}citB$, $\Delta ccpE^{Tn}citB$/p-*ccpE*, $\Delta isdC$, and $^{Tn}srtB$). The goal was to clarify the influence of CcpE on the production of IsdC, a vital component of the Isd system, which is attached to the cell wall by sortase B (SrtB) [33]. All the strains, except for $\Delta isdC$ and $^{Tn}srtB$ mutants, generated three principal IsdC-reactive bands (two with slower migration rates and one with a faster migration rate) when cultured in the presence of 1 mM DIP for 5 hours in TSB medium (Fig 4E). The two slower bands probably represent IsdC precursors (P1 and P2), while the faster band likely corresponds to the mature form of IsdC (Fig 4E). We found that deletion of either *citZ* or *ccpE* in the $^{Tn}citB$ mutant reduced the production of IsdC (Figs 4E and S7B).

## Citrate accumulation and activation of CcpE are triggered by iron starvation

Since CcpE is observed to bind to the promoter region and stimulates the expression of multiple Fur-regulated genes (S6 Fig), we hypothesize that citrate could serve as a crucial mediator linking central carbon metabolism and iron homeostasis. Indeed, it was noted that treatment

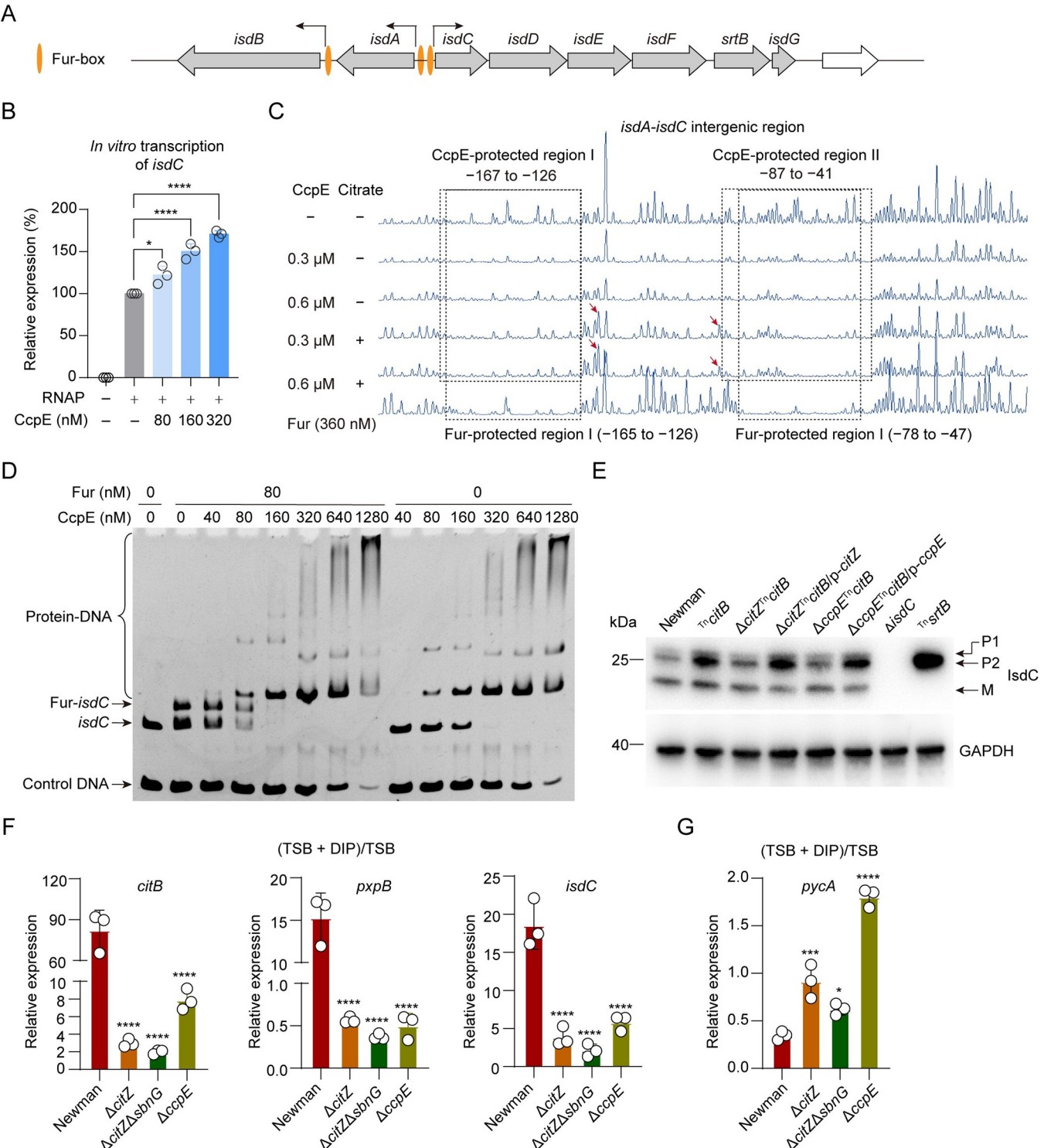

**Fig 4. CcpE competes with Fur for binding to the promoter region of *isdC*, and iron starvation activates CcpE.** (A) An illustration shows the genetic organization of the *isd* locus and the location of Fur-box. (B) Effect of CcpE on the *in vitro* transcription of *isdC* in the presence of sodium citrate (20 mM). Data reported as mean ± SD of *n* = 3 independent experiments, *$p < 0.05$, ****$p < 0.0001$ by One-way ANOVA with Dunnett test. (C) DNase I footprinting results of CcpE- and Fur-protected regions for the intergenic region between *isdA* and *isdC*. Protected regions (relative to the start codon of *isdC*) were framed with dashed rectangles, as indicated. Red arrows showed the hypersensitive sites. (D) EMSA experiments illustrating the impact of CcpE on the protein-DNA complexes formed by the Fur protein with the *isdC* promoter DNA (Fur-*isdC*). (E) Representative image of Western blotting for IsdC. Glyceraldehyde 3-phosphate dehydrogenase (GAPDH) was used as a loading control. Strains harbor either an empty plasmid control or p-*citZ* or p-*ccpE*, as indicated. (F and

G) Effect of DIP treatment on the expression of *citB*, *pxpB*, *isdC* (in F), and *pycA* (in G) in wild-type Newman, Δ*citZ*, Δ*citZ*Δ*sbnG*, and Δ*ccpE* cultured in TSB (without glucose) for about 4.5 h. Data presented with mean ± SD from $n = 3$ biological replicates, $* \, p < 0.05$, $*** \, p < 0.001$, $**** \, p < 0.0001$ by One-way ANOVA with Dunnett test, using the WT Newman group as the control.

with the DIP (S7C Fig) resulted in a 4-fold increase in the intracellular citrate level of the WT Newman strain (S7D Fig). In contrast, the $^{Tn}$*citB* mutant did not exhibit any significant alteration (S7D Fig). Thus, the DIP-induced citrate accumulation is potentially due to the inhibition of the CitB.

Upon DIP treatment, an 82.2% decrease in aconitase activity was observed in the WT strain of *S. aureus* Newman (S7E Fig). Using RT-qPCR, we detected elevated transcription levels of the *citB* gene in different growth phases of *S. aureus* WT Newman upon DIP treatment (S7F Fig), which implies activation of CcpE [12]. In addition, without DIP treatment, the maximum transcript level of *citB* occurred during the post-exponential growth phase, whereas in the presence of DIP, there was at least a 6-hour delay in the appearance of maximum *citB* transcript level during stationary growth phase (S7F Fig). Furthermore, we observed that the effects of DIP treatment on the expression of *citB*, *pxpB*, *isdC*, and *pycA* were largely mitigated by the deletion of either *citZ* or *ccpE* (Fig 4F and 4G). These findings suggest that iron deprivation results in citrate accumulation and the subsequent activation of CcpE.

However, the relationship between iron homeostasis and citrate accumulation in *S. aureus* appears to be intricate and multifaceted. Fur not only binds to the promoter and triggers *citB* expression *in vitro* and *in vivo* (S8A–S8E Fig), but it also represses IsrR, an sRNA that in turn downregulates CitB and CcpE through post-transcriptional control mechanisms [34]. Furthermore, the revelation that Fur serves an activator for *citB* aligns with earlier findings in *E. coli* [35], reinforcing the concept that Fur links iron transport and utilization enzymes with negative-feedback loop pairs for iron homeostasis [35].

## Citrate inhibits the transcriptional regulatory activity of Fur

Considering that citrate is known to sequester divalent cations [36,37], including $Fe^{2+}$ and $Mn^{2+}$, which are used by Fur to bind its target DNA [38,39], we performed an EMSA test to investigate whether citrate could influence the transcriptional regulatory function of Fur. In the assay, the purified recombinant Fur protein was exposed to *isdC* promoter DNA in the presence of $Mn^{2+}$ as a divalent metal, substituting for $Fe^{2+}$ as it is more stable in an aerobic environment and has the same coordination geometry [38,39]. The EMSA results demonstrated that citrate inhibits the binding of recombinant Fur to *isdC* promoter DNA in a dose-dependent way (Fig 5A). Furthermore, the addition of $Mn^{2+}$ into the EMSA binding mixtures, containing 20 mM citrate, resulted in a dose-dependent increase in the binding of Fur to the *isdC* promoter DNA (Fig 5B). We demonstrated through *in vitro* transcription assays that citrate has the ability to alleviate the inhibition of *isdC* transcription by the recombinant Fur protein (Fig 5C). Additionally, we observed that the deletion of *citZ* had a greater influence on the expression of Fur-regulated genes, including *sirA* and *sbnA* (Fig 5D), and to a lesser extent on *isdC* and *isdA* (Fig 5E), in the $^{Tn}$*citB* mutant in comparison to the *ccpE* deletion. These findings suggest that citrate can modulate the transcriptional regulatory activity of Fur in *S. aureus*.

## Citrate and CcpE can modulate the pathogenicity of *S. aureus* in a TCA-independent manner

To investigate the impact of citrate accumulation on the pathogenicity of *S. aureus*, we exposed WT *S. aureus* Newman and Δ*citZ*, $^{Tn}$*citB*, and Δ*citZ* $^{Tn}$*citB* mutant strains to a mouse model of

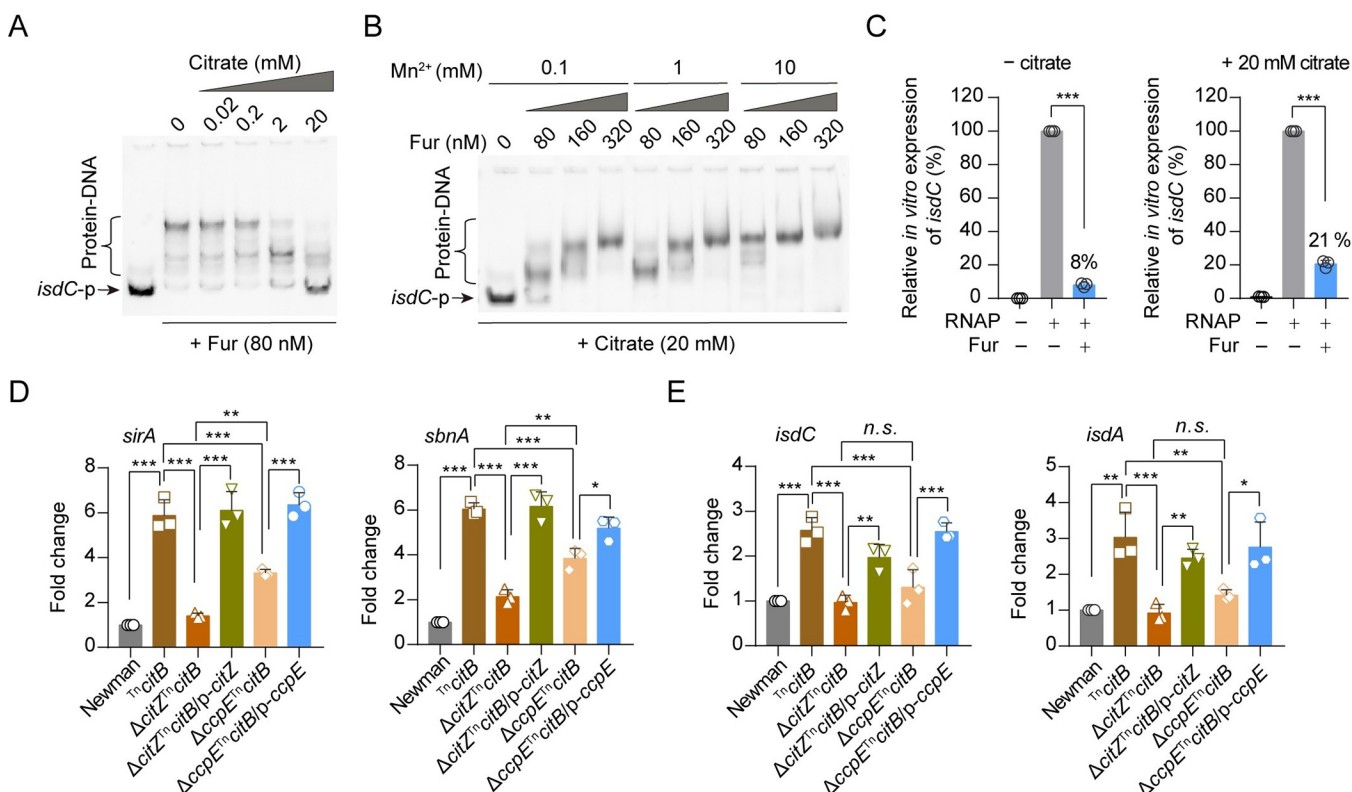

**Fig 5. Citrate inhibits the regulatory activity of Fur.** (A) EMSA showing the impact of citrate on the DNA binding ability of Fur. (B) EMSA showing the impact of $Mn^{2+}$ on the DNA binding ability of Fur in the presence of 20 mM citrate. (C) Effect of citrate on the transcriptional regulatory activity of Fur *in vitro*. Data presented with mean ± SD of $n = 3$ independent experiments. ***$p < 0.001$, by one sample *t*-test. (D and E) RT-qPCR analysis of *sirA*, *sbnA*, *isdC*, and *isdA* transcripts in *S. aureus* strains cultured in TSB supplemented with 1 mM DIP for 5 h. *n.s.*, $p > 0.05$, *$p < 0.05$, **$p < 0.01$, ***$p < 0.001$ by One-way ANOVA with Tukey test. Data represents the mean ± SD from $n = 3$ biological replicates.

bacteremia and analyzed bacterial survival in the host organs. Compared to the WT Newman strain, both Δ*citZ* and the citrate-accumulating <sup>Tn</sup>*citB* mutant exhibited statistically significant decreases in bacterial loads in the kidney or the hearts of infected mice (Fig 6A). This supports the idea that the TCA cycle regulates the pathogenicity of *S. aureus* [40,41]. Interestingly, we observed that the Δ*citZ*<sup>Tn</sup>*citB* mutant displayed approximately a 0.9-log increase in CFU bacterial loads in the kidneys and hearts of the infected mice compared to the <sup>Tn</sup>*citB* mutant, suggesting that intracellular citrate accumulation contributes to the decreased pathogenicity of the <sup>Tn</sup>*citB* mutant (Fig 6A). Furthermore, deleting *ccpE* also increased the bacterial loads of the <sup>Tn</sup>*citB* mutant in the kidneys (~0.9 log10 CFU/organ) in a statistically significant manner (Fig 6B). Additionally, although the increase was not statistically significant, we observed a rise (~1.5 log10 CFU/organ) in the bacterial load in the hearts for the <sup>Tn</sup>*citB* mutant upon *ccpE* deletion (Fig 6B). These results indicate that citrate and CcpE have the ability to regulate the pathogenicity of *S. aureus* in the <sup>Tn</sup>*citB* mutant.

To further investigate how *ccpE* modulates the pathogenicity of *S. aureus*, we conducted mouse infection experiments again using WT *S. aureus* Newman and Δ*pycA*, Δ*ccpE*, and Δ*pycA*Δ*ccpE* mutant strains. We observed a significant decrease in virulence of the *S. aureus* Newman strain upon deletion of *pycA*, which is consistent with previous studies [11,30]. The removal of *ccpE* resulted in a statistically significant increase in bacterial loads of the Δ*pycA* mutant in the kidneys and hearts (Fig 6C), indicating that CcpE can modulate the pathogenicity of *S. aureus* Newman in a manner independent of *pycA*. However, the role of citrate in the

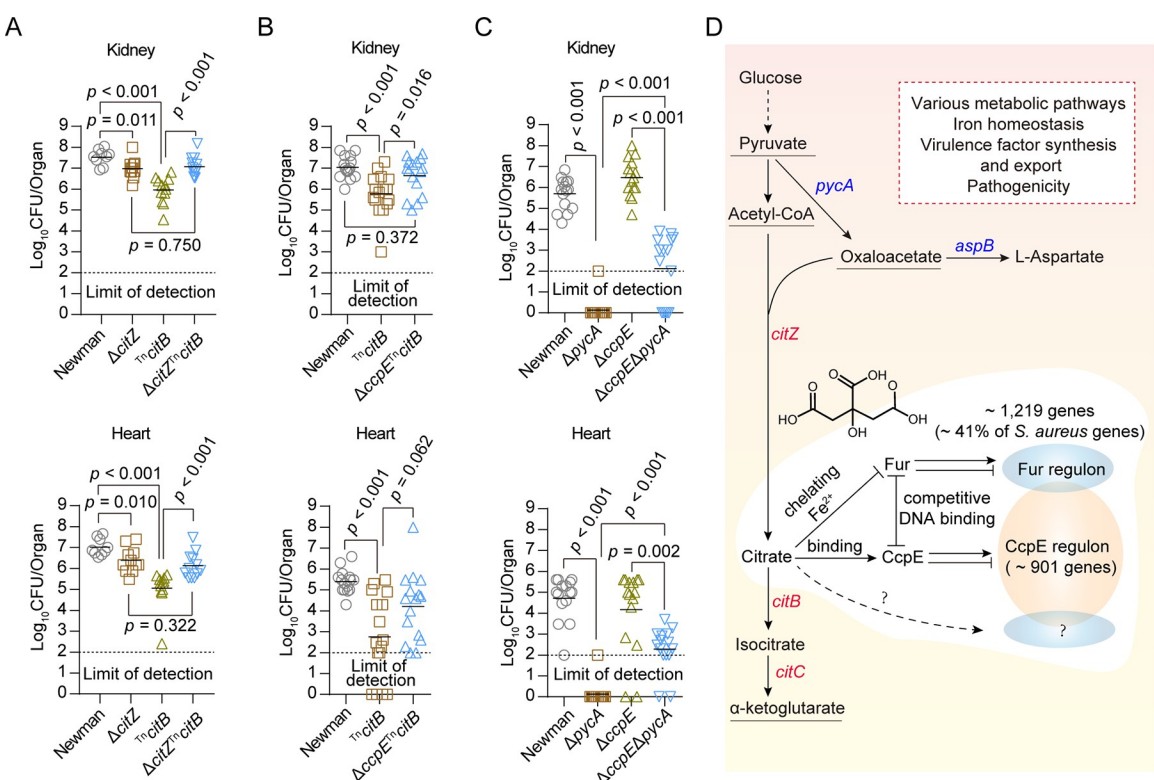

**Fig 6. Virulence of *S. aureus* strains and a proposed model for the regulation by citrate.** (A to C) *S. aureus* colonization in a mouse bacteremia model. Female BALB/c mice were infected retro-orbitally with $4 \times 10^6$ CFU (in A), $3 \times 10^6$ CFU (in B), or $1 \times 10^6$ CFU (in C) of *S. aureus*. Each symbol represents data from one experimental animal. Horizontal bars indicate observation means, and dashed lines mark limits of detection. Statistical analysis was performed using Mann-Whitney test. (D) A proposal for how citrate regulates genes in *S. aureus*. Arrows indicate activation or generation, blunted lines indicate inhibition, solid lines indicate a direct influence or connection, and dotted lines indicate indirect influence. Genes expressed in blue or red indicate those whose expression is directly inhibited or activated by CcpE.

pathogenicity of *S. aureus* may be underestimated in the current infection model, as *S. aureus* has developed its Isd system with a heightened capacity to bind hemoglobin derived from humans compared to other mammals [42] and citrate influences the expression of *Isd* genes (S6 Fig). The regulatory effect of citrate or CcpE on the pathogenicity of *S. aureus* seems to be multifaceted, and additional studies are warranted.

## Discussion

Bacterial pathogens have developed intricate mechanisms to assimilate nutrients and regulate flux through metabolic pathways to thrive and establish infection in their host [18,30,43–45]. In this study, we revealed that citrate, the first intermediate in the TCA cycle, plays a crucial role in the transcriptional regulation of *S. aureus* genes by modulating the transcription factor CcpE and Fur (Fig 6D). In addition, we found that citrate regulates the pathogenic potential of *S. aureus*, independent of its canonical role as a TCA cycle intermediate (Fig 6A).

In organism, central carbon metabolism (which includes glycolysis, the pentose phosphate pathway, and the TCA cycle) utilizes a complex series of enzymatic steps to convert sugars into 12 branchpoint metabolites (precursor compounds) necessary to synthesize macromolecules [32] (S5 Fig). Citrate, among the carbon metabolites, is a non-precursor molecule and is primarily recognized as the first intermediate of the TCA cycle [26,32] (S5 Fig). In mammalian

cells, citrate also function as a vital regulator by inhibiting and inducing strategic enzymes situated at the entrance and/or at the exit of glycolysis, gluconeogenesis, fatty acids synthesis, and the TCA cycle [46]. For example, citrate plays a crucial role in inhibiting the activity of 6-phosphofructo-1-kinase (PFK) and thereby regulating glycolytic flux [46]. However, citrate seems to do not affect bacterial PFK [47], and there is currently less available information on the regulatory functions of citrate in prokaryotes cells.

Our study uncovered that changes in intracellular citrate levels can influence the expression of a vast array of *S. aureus* genes (Fig 1F), encompassing over 40% of the genome of this significant human pathogen. Although the precise mechanisms through which citrate modulates the *S. aureus* transcriptome remains undefined, we have shown that the regulatory effect of citrate is predominantly mediated by CcpE (Fig 6D). Additionally, citrate can also influence Fur repressor activity by directly chelating intracellular iron, which is essential for Fur repressor activity [39,48], and by co-activating CcpE, which competitively displaces Fur (Fig 4C and 4D). However, it is important to note that both the *citB* and Δ*citZ*$^{Tn}$*citB* mutants exhibit a partial impairment of the TCA cycle (as evidenced by post-exponential growth defects in Fig 1C), introducing complexity in determining whether the observed effects are solely attributed to citrate or influenced by impaired TCA cycle function or growth-related factors.

Our IDAP-Seq experiments demonstrated CcpE's binding preference to promoters of genes (or operons) encoding crucial enzymes located at the entrance of various metabolic pathways, including gluconeogenesis (*pycA*), fatty acids synthesis (*accBC*), and the TCA cycle (*citZC*), as well as aconitase (*citB*), which catalyzes the second step of the TCA cycle in *S. aureus* (Figs 2 and S4, and S3 Table). Indeed, CcpE functions as a negative regulator of *pycA* (Fig 3), while acting as a positive regulator of *accBC* and *citB*, and to a lesser extent, *citZC* (S4 Fig). However, the mechanisms underlying the dual role of CcpE as an activator for some genes and a repressor for others remain to be defined. Nonetheless, these observations emphasize that CcpE may play a crucial role in regulating the central carbon metabolism of *S. aureus*, as changes in the expression of these CcpE-targeted genes are predicted to impact the levels of four branchpoint metabolites: pyruvate, oxaloacetate, acetyl-CoA, and α-ketoglutarate (S5 Fig) [32].

Citrate regulates the expression of Fur-regulated genes (S6 Fig) and those involved in iron-sulfur cluster biogenesis (i.e., *sufCDSUB* operon) (S2 Table), but the impact of citrate on the iron hemostasis of *S. aureus* may extend far beyond gene expression regulation. Indeed, citrate serves as a central component of two siderophores (i.e., staphyloferrin A and staphyloferrin B) that promote the growth of *S. aureus* under iron-restriction conditions [40]. Interestingly, TCA derived citrate is only necessary for staphyloferrin A production, not for staphyloferrin B production [40]. It also has been reported that different types of bacteria, both Gram-positive and Gram-negative species, use citrate or citrate-based siderophore for iron scavenging from the cell environment [49]. Moreover, it is important to emphasize that iron is an essential cofactor that enables a wide range of key metabolic enzyme activities, and the TCA cycle contains at least two iron-requiring enzymes, namely aconitase and succinate dehydrogenase [50,51]. Studies have reported that the de-repression of the TCA cycle leads to a considerable demand for iron during the post-exponential growth phase in *S. aureus* [50,51].

The metabolism of glucose plays a crucial role in the *S. aureus* pathogenicity [9,52]. Studies have indicated that elevated glucose levels in the hyperglycemic abscesses can significantly boost the virulence potential *S. aureus*, leading to more severe infection outcomes [53]. The utilization of glucose by *S. aureus* can occur through two main metabolic pathways: fermentation and respiration [18,54,55]. These pathways appear to play a significant role in determining the levels of citrate within the bacterium [18,54,55]. During fermentation, *S. aureus* metabolizes glucose to produce energy in the absence of oxygen, resulting in the formation of

fermentation byproducts such as lactate, ethanol, and acetate [54,55]. On the other hand, when oxygen is present, *S. aureus* metabolizes glucose and generates acetate as the primary byproduct due to the repression of TCA by the CcpA protein [18]. After glucose is consumed, the accumulated acetate is further metabolized through the citric acid cycle and oxidative phosphorylation [18,54,55]. We observed that deletion of *ccpE* in the [Tn]*citB* mutant led to markedly increased expression of genes, such as *butA*, *NWMN_2459*, *lctP1*, *NWMN_1315*, *NWMN_2026*, *adh1*, *pflB*, and *fdhA*, associated with fermentation pathways [56,57] (S5 Fig). Additionally, in the pathway from pyruvate to L-leucine through the 2,3-Dihydroxy-3-methyl-butanoate in the [Tn]*citB* mutant, the levels of L-leucine and 2,3-Dihydroxy-3-methylbutanoate were significantly upregulated upon *ccpE* deletion, while glycolytic intermediates like glycerate-3P, glycerate-2P, and phosphoenolpyruvate (PEP) exhibited a decrease in abundance (S5 Fig). Furthermore, we also observed that level of L-alanine, a fermentation product, was increased in the [Tn]*citB* mutant by the deletion of *ccpE* (S5 Fig). These results, combined with CcpE's role as an activator of the TCA cycle [12,25], suggests that CcpE may play a pivotal role in coordinating the metabolic transition from fermentation to respiration.

Interestingly, in addition to *pycA*, CcpE also binds to the promoter regions and inhibits the transcription of *zwf* and *glmS* (S5 Fig). In bacteria, the protein product of *zwf* diverts glucose-6P from glycolysis into the pentose phosphate pathway, while that of *glmS* catalyzes the conversion of fructose-6P to GlcN-6P [58] (S5 Fig). These observations are also consistent with the finding that deletion of *ccpE* decreased the levels of glycolytic intermediates like glycerate-3P, glycerate-2P, and phosphoenolpyruvate (PEP) in the [Tn]*citB* mutant (S5 Fig), suggesting that activation of CcpE may help to direct the carbon flux to the glycolytic pathway. Moreover, the impact of CcpE on central carbon metabolism may be more profound than previously thought. CcpE can bind to promoter regions and/or influence the expression of genes involved in the formation (or consumption) of almost all 12 branchpoint metabolites (S5 Fig), which serve as exit points for carbon from the central carbon metabolism network in bacteria [32] (S5 Fig).

In summary, our research highlights the crucial non-metabolic roles of citrate in regulating gene expression (Fig 6D). Citrate acts as a key link between multiple cellular processes, coordinating the glycolytic pathway, TCA cycle, and iron uptake at the transcriptional level in *S. aureus* (Figs 6D and S5). There is still much uncover regarding the regulation and implications of "citrate signaling" in cells. A more thorough comprehension of metabolic signaling and regulation could offer substantial for exploring and addressing various health and disease-related concerns.

## Materials and methods

### Ethics statement

Animal experiments were carried out strictly in accordance with the regulations of the Administration of Affairs Concerning Experimental Animals approved by State Council of People's Republic of China (11-14-1988). Animal studies were approved and supervised by the Institutional Animal Care and Use Committee (IACUC) of Shanghai Public Health Clinical Centre. Laboratory animal usage license was certified by Shanghai Committee of Science and Technology, the certification number is SYXK-HU-2010-0098.

### Bacterial strains, plasmids, and culture conditions

Bacterial strains and plasmids used in this study were listed in S5 Table. Unless otherwise described, *Escherichia coli* was grown in Luria Bertani (LB) broth or on LA (LB agar) plates. *S. aureus* strains were cultivated in Tryptone Soya Broth (TSB) (Oxoid, Catalog no. CM0129) or

on TSA plate (Oxoid, Catalog no. CM0131). When necessary, antibiotics were used at the following concentrations for maintaining plasmids: for *E. coli*, 100 μg/mL carbenicillin or 50 μg/mL kanamycin; for *S. aureus*, 10 μg/mL erythromycin or 10 μg/mL chloramphenicol. For routine cultivating, all of the strains were grown aerobically at 37˚C with shaking at 250 rpm.

## Construction of *S. aureus* null or transposon insertion mutants

For creating *S. aureus* gene null mutants, an allelic replacement strategy was employed [59]. Generally, polymerase chain reactions (PCRs) were performed to amplify upstream sequences (~1 kb) as well as downstream sequences (~1 kb) of the targeted deletion gene. About 10 ng of both the upstream and downstream products were used as templates for the second round of overlapping PCR, resulting in the upstream + downstream fragment. This fragment was used for recombination with pKOR1, a plasmid which permits rapid cloning *via* lambda recombination and *ccdB* selection. Recombination reactions were performed using Gateway cloning kit (Invitrogen, Catalog no. 11789–020). The resulting products were subsequently transformed into *E. coli* DH5a competent cells (Yeasen Biotech. Co. Ltd., Catalog no. 11802ES80). Correct plasmids were electroporated transformed into *S. aureus* 4220 strain (restriction minus and modification plus) to modify plasmids. The modified plasmids were subsequently transformed into *S. aureus* Newman strain. Allelic replacement was performed and the null mutant alleles were verified by PCR and DNA sequencing.

For the deletion of *citZ*, the upstream fragment and downstream fragment of intended deletion was amplified from *S. aureus* Newman genomic DNA using primers *citZ*-up-F/*citZ*-up-R and *citZ*-down-F/*citZ*-down-R (S6 Table.), respectively. For the deletion of *ccpE*, four primer, *ccpE*-up-F/*ccpE*-up-R and *ccpE*-down-F/*ccpE*-down-R, were used. For the deletion of *sbnG*, primers *sbnG*-up-F/*sbnG*-up-R and *sbnG*-down-F/*sbnG*-down-R were used. For the deletion of *pycA*, primers *pycA*-up-F/*pycA*-up-R and *pycA*-down-F/*pycA*-down-R were used. For the deletion of *isdC*, primers *isdC*-up-F/*isdC*-up-R and *isdC*-down-F/*isdC*-down-R were used. For the deletion of *fur*, primers *fur*-up-F/*fur*-up-R and *fur*-down-F/*fur*-down-R were used.

JE2 ^Tn^*citB* (NE861, Catalog no. NR-47404) and JE2 ^Tn^*srtB* (NE1363, Catalog no. NR-47905) were acquired from the Nebraska transposon mutant library *via* the Network on Antimicrobial Resistance in *S. aureus* (http://www.narsa.net). The transposon was inserted 591 bp downstream the translational start site of *citB* (locus tag: SAUSA300_1246). For JE2 ^Tn^*srtB*, the transposon was inserted 260 bp downstream the translational start site of *srtB* (locus tag: SAUSA300_1034). Newman ^Tn^*citB* and Newman ^Tn^*srtB* mutant were generated by phage transduction [41] with Φ85 phage lysate from JE2 ^Tn^*citB* or JE2 ^Tn^*srtB*, respectively. To obtain Δ*citZ*^Tn^*citB* or Δ*ccpE*^Tn^*citB* double mutants, Δ*citZ* or Δ*ccpE* (either of each was generated by allelic replacement as described above) was used as the recipient, and transduction was performed similarly as described above. All of the mutants were confirmed by PCR and DNA sequencing.

## Construction of plasmids for gene complementation

To construct plasmids for gene complementation, DNA fragments covering the ribosome binding site and the whole coding regions were amplified from the genomic DNA of Newman, and subsequently cloned into pYJ335 pre-digested with *Eco*RV. To create plasmid p-*citB*, a ~2.7 kb DNA fragment containing *citB* was amplified from *S. aureus* Newman genomic DNA with primers *citB*-F and *citB*-R. To create plasmid p-*citZ*, a~1.2 kb DNA fragment containing *citZ* was amplified from *S. aureus* Newman genomic DNA with primers *citZ*-F and *citZ*-R. To create plasmid p-*sbnG*, a~0.8 kb DNA fragment containing *sbnG* was amplified from *S. aureus* Newman genomic DNA with primers *sbnG*-F and *sbnG*-R. To create plasmid p-*pycA*, a~3.5 kb

DNA fragment containing *pycA* was amplified from *S. aureus* Newman genomic DNA with primers *pycA*-F and *pycA*-R. To create plasmid p-*fur*, a ~0.5 kb DNA fragment containing *fur* was amplified from *S. aureus* Newman genomic DNA with primers *fur*-F and *fur*-R. Clones with target gene downstream of the tetracycline-inducible *xyl/tetO* promoter were screened by PCR and confirmed by DNA sequencing.

### Construction of plasmids for protein expression

The coding regions of CcpE, Sigma A, and Fur were amplified from the genomic DNA of *S. aureus* Newman with primers pET28a::*ccpE*-F/pET28a::*ccpE*-R, pET28a::*sigma* A-F/pET28a::*sigma* A-R, and pET28a-*sumo*::*fur*-F/pET28a-*sumo*::*fur*-R, respectively. Amplified fragments were digested with *Bam*HI and *Xho*I (for pET28a::*ccpE* and pET28a::*sigma A*) or *Kpn*I and *Hind*III (for pET28a-*sumo*::*fur*), and inserted into pET28a or pET28a-*sumo* digested with the same pair of restriction enzymes to generate protein expression plasmids. All of the constructs were confirmed by PCR and DNA sequencing.

### Construction of plasmids for *pycA*-*lacZ* transcriptional fusions

For constructing *pycA*-*lacZ*-reporter plasmid, the *pycA* promoter region (-554 to +21 relative to the translational start codon) was PCR amplified from the genomic DNA of Newman with primers *pycA*-*lacZ*-F/*pycA*-*lacZ*-R and digested with *Eco*RI and *Kpn*I. The resulting fragment was cloned into pCL-*lacZ*, which carrying a promoterless *lacZ* reporter gene, generating a promoter-*lacZ* reporter fusion (pCL-*pycA*-*lacZ*). For site-directed mutagenesis of *pycA*-*lacZ*-reporter plasmid, PCR was performed with primers *pycA* (G-$N_{11}$-G)-*lacZ*-F/*pycA* (G-$N_{11}$-G)-*lacZ*-R, using pCL-*pycA*-*lacZ* as the template. After amplification, the template was digested with *Dpn* I at 37°C for 3 h. The digested mixtures were then transformed into chemically-competent *E. coli* DH5α cells. The presence of the desired mutations was confirmed through DNA sequencing.

### Metabolomic analysis

Overnight cultures of *S. aureus* Newman, $^{Tn}citB$, $\Delta citZ^{Tn}citB$, and $\Delta ccpE^{Tn}citB$ strains were adjusted to $OD_{600} = 5.0$ and transferred 1:100 into 10 mL fresh TSB medium, with a tube volume to medium volume = 5:1. Six biological replicates were included in each group and all of the sub-cultures were cultivated at 37°C with shaking at 250 rpm for 12 h. Bacterial cultures were flash frozen in liquid nitrogen and stored at -80°C until use. For LC-MS/MS analysis, samples were thawed at 4°C and washed twice with pre-cold PBS buffer. Subsequently, 1 mL methanol/acetonitrile/$H_2O$ (2:2:1, v/v/v) was added to pellets, followed by vortex for 60s. The mixtures were further sonicated at 4°C with a total time of 60 min, and vortex was performed between two intervals. Sonicated samples were stored at -20°C for 1 h to precipitate proteins and proteins were removed by centrifugated at 14,000 g, 4°C for 20 min. Supernatant was then dried by lyophilization and re-dissolved in 100 μl acetonitrile/water (1:1, v/v) solvent. To monitor the repeatability and stability of instrument, quality control (QC) sample was prepared by pooling 10 μL of each sample and analyzed together with other samples.

LC-MS/MS analysis was performed using an Ultra-High-Performance Liquid Chromatography (UHPLC, 1290 Infinity LC, Agilent Technologies) coupled to a quadrupole time of flight mass spectrometry (AB Sciex TripleTOF 6600). For Hydrophilic Interaction Liquid Chromatography (HILIC) separation, samples were analyzed by a 100 mm×2.1 mm ACQUIY UPLC BEH 1.7 μm column (Waters, Ireland) with a flow rate of 0.3 ml/min and column temperature of 25°C. In both Electrospray Ionization (ESI) negative and positive modes, the mobile phase was A = 25 mM ammonium hydroxide and 25 mM ammonium acetate in water and

B = acetonitrile. The gradient was 95% B for 1 min and was linearly reduced to 65% in 13 min, followed by reducing to 40% in 2 min and kept for 2 min, and then increased to 95% in 0.1 min. 5 min of re-equilibration period was employed before analysis for the next sample. ESI source conditions were set as the following: Ion Souce Gas1 (Gas1) as 60, Ion Source Gas2 (Gas2) as 60, curtain gas (CUR) as 30, source temperature: 600°C, Ion Spray Voltage Floating (ISVF) ±5500V. Instrument was set to acquire over the m/z range of 60–1000 Da in MS only acquisition, and TOF MS accumulation time was set at 0.20 s/spectra. In auto MS/MS acquisition, a m/z range of 25–1000 Da and 0.05 s/spectra for accumulation time of product ion scan were employed. The product ion scan is acquired using information dependent acquisition (IDA) with high sensitivity mode selected. Collision energy (CE) was fixed at 35 V±15 eV and declustering potential (DP) was set as ± 60 V.

For data processing, raw MS data (wiff. scan files) was converted to MzXML files via ProteoWizard MSConvert and processed using XCMS for retention time correction, feature detection and alignment, with default parameter settings. Identification of metabolites was performed by comparing the accuracy mass (<25 ppm), and MS/MS spectra with the laboratory's self-built commercial database (Shanghai Applied Protein Technology Co., Ltd.). In the extracted ion features, only the variables having more than 50% of the non-zero measurement values in at least one group were kept. For the multivariate statistical analysis, the MetaboAnalyst (www.metaboanalyst.ca) web-based system was used. After the data was preprocessed by Pareto-scaling, multi-dimensional statistical analysis, including unsupervised principal component analysis (PCA) and orthogonal partial least-squares discriminant analysis (OPLS-DA) were performed. The first principal component of the variable importance in the projection (VIP) was obtained from OPLS-DA to refine the analysis. The metabolites with VIP values ≥1 and the variables assessed by Student's two-tailed *t*-test with $p$-value < 0.05 were considered as significantly differential metabolites. Differential metabolites were further identified and validated by databases including Kyoto Encyclopedia of Genes and Genomes (KEGG, http://www.kegg.jp/) and Human Metabolome Database. Fold-change values of metabolites were calculated by comparing the mean value between each group. Metabolomic analysis was performed with the assistance of Shanghai Applied Protein Technology Co. Ltd. All metabolomic data (six biological replicates for each sample) has been submitted to the database of MetaboLights (https://www.ebi.ac.uk/metabolights/) with the accession numbers MTBLS4276.

### RNA-seq and data analysis

Overnight cultures of *S. aureus* Newman, $^{Tn}citB$, $\Delta citZ^{Tn}citB$, and $\Delta ccpE^{Tn}citB$ strains were adjusted to $OD_{600}$ = 5.0 and transferred to 10 mL of fresh TSB medium at a ratio of 1:100. The tube volume to medium volume ratio was of 5:1. Three biological replicates were included in each group and all of the sub-cultures were cultivated at 37°C with shaking at 250 rpm for 12 h. For RNA extraction, cultures were harvested by centrifugation at 4°C. After resuspended in 1 mL pre-cold sterile 1 × TE buffer, pellets were homogenized by a homogenizer (Next Advance Bullet Blunder Storm 24). Supernatant containing RNA were collected after centrifugation at 14,000 g, 4°C for 4 min. Total RNA was extracted by a QIAGEN RNeasy extraction kit (Catalog no. 74104) following the manufacture's recommendations. Ribosomal RNA removal, cDNA library construction and paired-end sequencing with Illumina HiSeq 2000 were performed by Guangdong Magigene Biotechnology Co. Ltd. Differential expressed genes (DEGs) were detected by the edgeR software package with a fold change ≥ 2 and an $p$-value < 0.05 (edgeR, exact test). All RNA-seq data (triplicate biological replicates for each sample) has been submitted to the NCBI Sequence Read Archive (SRA, https://ncbi.nlm.nih.gov/sra/) under the

BioProject accession number PRJNA797550, with the BioSample accession numbers SAMN25010747 to SAMN25010758.

## Joint pathway analysis

The transcriptomics-metabolomics coupled analysis was conducted utilizing the Joint Pathway Analysis module of MetaboAnalyst 6.0 [22], with the *S. aureus* N315 strain serving as the model organism. Genes and metabolites exhibiting significant regulation (gene with ≥2-fold change and $p<0.05$, and metabolites with VIP≥1 and $p<0.05$) were selected for Joint Pathway Analysis. This analysis was based on all pathways, including both metabolic pathways and gene-only pathways (i.e. regulatory pathways), with the enrichment analysis grounded in the hypergeometric test. The calculation of $p$-value was achieved through an integrated approach, wherein genes and metabolites are pooled into a single query and used to perform enrichment analysis within their "pooled universe". For pathways with hits from only one input type, p values calculated from their individual universe will be used. Pathway impact values were determined based on pathway topology measures, which evaluates the potential importance of a particular molecule based on its position within a pathway. Degree centrality was employed as the default topology measurement method for this analysis.

## Overexpression and purification of recombinant CcpE, SigA, and Fur proteins

The proteins were expressed in *E. coli* strain BL21(DE3) and purified by Ni-NTA affinity chromatography. Generally, overnight cultures were sub-cultured 1:100 into 1L of fresh LB medium supplemented with 50 μg/mL kanamycin and grown to an $OD_{600}$ of 0.6–0.8. Expression of CcpE with a N-terminal fused His tag (6×His-CcpE) or Sigma A with a N-terminal fused His tag (6×His-SigA) was induced with 0.5 mM isopropyl-β-D-thiogalactoside (IPTG) overnight at 16°C or 25°C, respectively, with shaking at 220 rpm. Expression of Fur with a N-terminal fused Sumo tag (Sumo-Fur) was induced with 0.5 mM IPTG at 30°C for 4 h. Cells were harvested and bacteria pellets were resuspended in 50 mL buffer A (for 6×His-CcpE: 20 mM Tris-HCl, pH 8.0; 300 mM NaCl; 10% glycerol; 5 mM 2-mercaptoethanol; for 6×His-SigA: 50 mM Tris-HCl, pH 8.0; 300 mM NaCl; 10% glycerol; for Sumo-Fur: 50 mM Tris-HCl, pH 7.5; 300 mM NaCl; 10% glycerol; 5 mM 2-mercaptoethanol). Bacterial suspensions were lysed at 4°C by high pressure homogenization for about 5 min. Lysates were centrifuged at 15,000 g for 25 min, the resulting supernatants were loaded onto a His Trap HP 5-ml column (GE Healthcare, Catalog no. 17-5248-01) pre-equilibrated with buffer A. His-tagged proteins were purified with an AKTA pure system. Briefly, 40 mL 10% buffer B (buffer A plus 400 mM imidazole) was used to elute most of the non-specific binding proteins, followed by step elution with 10–100% buffer B at a flow rate of 2 mL/min for 15 column volumes (CVs). Purified proteins were analyzed by SDS-PAGE with Coomassie blue staining. Fractions containing the targeted proteins were pooled.

For 6×His-SigA and Sumo-Fur, pooled proteins were dialyzed at 4°C with 2 L buffer C (for His-SigA: 50 mM Tris-HCl, pH 8.0; 150 mM NaCl; 10% glycerol; for Sumo-Fur: 50 mM Tris-HCl, pH 7.5; 150 mM NaCl; 10% glycerol; 5 mM 2-mercaptoethanol;) for 6 h~8 h to remove most of the imidazole. After dialysis was complete, purified proteins were sub-packaged and stored at -80°C until use.

For 6×His-CcpE, proteins were concentrated and filtered, followed by removing imidazole with a Hi Trap Desalting column (GE healthcare, Catalog no. 17-1408-01) using buffer A. The resulting 6×His-CcpE was stored and used for electrophoretic mobility shift and DNase I footprinting assays. 6×His-CcpE used for *in vitro* transcription were further purified by Superdex 200 (10/300) gel-filtration chromatography (GE Healthcare, Catalog no. 28-9909-44) with buffer A, after Ni-column affinity purification.

### *In vitro* DNA affinity purification and sequencing (IDAP-Seq)

The genomic DNA of *S. aureus* Newman was extracted using a DNA purification kit (Tiangen, Catalog no. DP302-02) after lysis with lysostaphin. DNase free RNase A was added to a final concentration of 20 µg/mL to digest RNA at 37°C for 1h. After digestion, genomic DNA was sheared by sonication (SCIENTZ, Ultrasonic homogenizer, JY92-IIDN) to a fragment size range of 300 to 600 bp with a total working time of 8 min (one second on and one second off on ice, amplitude transformer 2, 10% power). Sheared DNA was recovered with a Cycle-Pure kit (Omega, Catalog no. D6492-02). A total volume of 300 µl, including 2.4 µg recovered DNA, 140 nM purified 6×His-CcpE, and 1×binding buffer (20 mM Tris-HCl, pH 7.4; 50 mM KCl; 20 mM $MgCl_2$; 1 mM EDTA; 5% NP-40; 5% glycerol; 1 mM DTT; 20 mM citrate), was used to perform *in vitro* DNA affinity purification. The binding mixtures were incubated at room temperature for 15 min. During this time, Ni-NTA agarose resin (Yeasen Biotech. Co.Ltd., Catalog no. 20502ES10) was washed twice with $ddH_2O$ and pre-equilibrated twice with 300 µL 1×binding buffer. Subsequently, the binding mixtures were added to the pre-equilibrated Ni-NTA agarose resin and the whole mixtures were incubated at 4°C with gentle rotation for 30 min. After that, the resin was washed four times with 300 µL 1×binding buffer and followed by eluted with 300 µL elution buffer (20 mM Tris-HCl, pH 7.4; 50 mM KCl; 20 mM $MgCl_2$; 1 mM EDTA; 5% NP-40; 5% glycerol; 1 mM DTT; 20 mM citrate; 500 mM imidazole) at room temperature for 30 min. Finally, the eluent was purified with a Cycle-Pure Kit (Omega, Catalog no. D6492-02).

Purified DNA samples were quantified by a Qubit 2.0 Fluorometer (Invitrogen) and qualified by Agilent Bioanalyszer 2100 (Agilent Technologies). Next generation sequencing library preparations were constructed according to the manufacturer's protocol (NEBNext II DNA library Prep kit for Illumina). For each sample, at least 10 ng *in vitro* affinity purified DNA or input DNA (sheared genomic DNA before *in vitro* DNA affinity purification) was used for library preparation. Final libraries were validated with Agilent 2100 Bioanalyzer and then sequenced with the HiSeq X system (Illumina). Each sample generated about 20 million reads and technical sequences, including adapters, PCR primers, and quality of bases lower than 20 bp were removed by Cutadapt (version 1.9.1) to obtain high quality clean data. Clean reads were mapped to the *S. aureus* Newman genomes using Bowtie (version 2.2.6) [60]. Enriched peaks were identified by Model-based Analysis of Chip-seq 2 (MACS2) software [61]. Peaks identified by MACS2 with an adjusted score of False Discovery Rate (FDR) corrected *p*-value (i.e., $-\log_{10}$ q-value) of at least 50 was set to ensure that a high-quality peak annotation was obtained. The original IDAP-Seq data files were deposited to the NCBI Sequence Read Archive (SRA, https://ncbi.nlm.nih.gov/sra/) under the BioProject accession number PRJNA796052, with the BioSample accession numbers SAMN24812581 to SAMN24812584.

### Multiple EM for Motif Elicitation (MEME) analysis

To identify the conserved DNA binding sites of 6×His-CcpE, all 107 peaks (fold enrichment $\geq 2$ and $-\log_{10}$ q value of $>50$) occurred at least twice in the three independent replicates were selected and used to find the conserved motif. For motif detection, sequences of ± 40 bp from the peak summit (81bp in total) were submitted for motif analysis by MEME (https://meme-suite.org/meme/tools/meme) with the minimum motif width set at thirteen, and other parameters set as default [62].

To find the conserved DNA binding sites of Fur, twenty-three potential Fur boxes, each comprising a 19 bp fragment, were download from the RegPrecise website (https://regprecise.lbl.gov/sites.jsp?regulog_id=670), which used *S. aureus* N315 as a model organism. These 23 Fur boxes distributed in the promoter regions of twenty genes or operons in total. Among

them, three Fur boxes were located in the promoter region of *ftnA* (*SA1709*), two Fur boxes were located in the promoter region of the *arlRS* operon (*SA1248-SA1246*), and the remaining eighteen Fur boxes were located in the promoter regions of *icuB* (*SA1983*), *isdH* (*SA1552*), *isdB* (*SA0976*), *isdCDEF-srtB-isdG* (*SA0978-SA0983*), *isdA* (*SA0977*), *fhuD2* (*SA2079*), *fhuABG* (*SA0602-SA0604*), *feoAB* (*SA2338-SA2337*), *efeOBU* (*SA0331-SA0333*), *citB* (*SA1184*), *yfmCDE* (*SA1979-SA1977*), *trxB* (*SA2162*), *tatAC* (*SA0335-SA0334*), *sstABCD* (*SA0688-SA0691*), *sirABC* (*SA0111-SA0109*), *sbnABCDEFGHI* (*SA0112-SA0120*), *sfnaABC* (*SA1982-SA1980*), *ntrA* (*SA0757*), respectively. These Fur boxes were aligned with the homologs from *S. aureus* Newman, and only one base pair of the *tatA* putative Fux box (ATTGATAATAATTATCGTT in Newman) is different from that of *S. aureus* N315 (ATTGATAATCATTATCGTT). Finally, the Fur boxes from *S. aureus* Newman were submitted for motif analysis by MEME with default settings.

## Electrophoretic Mobility Shift Assay (EMSA)

EMSA was performed with minor modifications following the previously described protocol [63,64]. Fragments used for EMSA were generated by PCR using the genomic DNA of Newman as a template. For the promoter of *isdC*, a 359 bp fragment covering nucleotide -252 to nucleotide +107 relative to the start codon of *isdC* was amplified with the primer *isdC*-EMSA-F/*isdC*-EMSA-R. For the promoter of *citB*, a 697 bp fragment covering nucleotide -624 to nucleotide +73 relative to the start codon of *citB* was amplified with the primer *citB*-EMSA-F/*citB*-EMSA-R. For the coding sequence of *murA*, a 1260 bp fragment covering nucleotide +3 to nucleotide +1262 relative to the start codon of *murA* was amplified with the primer *murA*-EMSA-F/*murA*-EMSA-R. This fragment was used as a control DNA fragment to detect the binding ability of Fur with the promoter of *citB*. All PCR products were purified with a Cycle-Pure kit (Omega, Catalog no. D6492-02).

To perform EMSA with Sumo-Fur, the following binding buffer was used: 10 mM Tris-borate, pH 8.0; 1 mM $MgCl_2$; 25 mM KCl; 100 µM $MnCl_2$; and 50 µg/mL BSA. Reaction mixtures were incubated at room temperature for 20 min and 2 µL 10×EMSA loading buffer (Beyotime, Catalog no. GS007) was added to the mixtures before electrophoresis. Electrophoresis was carried out with 5% non-denaturing polyacrylamide gel in 0.5×TB buffer (45 mM Tris, 45 mM borate, pH 8.0) plus 100 µM $MnCl_2$. To evaluate the impact of citrate on DNA binding affinity of Sumo-Fur, citrate was added to the binding buffer with indicated concentrations.

To test the competition of CcpE and Fur for their binding to the promoter of *isdC*, one protein was first added at a fixed concentration into the reaction buffer (10 mM Tris-borate, pH 8.0; 1 mM $MgCl_2$; 25 mM KCl; 100 µM $MnCl_2$; and 50 µg/mL BSA, 20 mM citrate) and incubation for 10 minutes at 25˚C, the other protein was then added at a gradually increased concentration in order to compete with the first one for the binding, after incubation at 25˚C for another 25 min, reaction products were loaded onto 4.5% non-denaturing polyacrylamide gel and followed the procedure as described above. A 200-bp DAN fragment covering nucleotide -127 to nucleotide +73 relative to the start codon of *citB* was amplified with the primer *citB*-EMSA-F2/*citB*-EMSA-R, and was used as the control DNA. The Fur proteins were obtained by treating Sumo-Fur with ULP1 (invitrogen) at 4˚C overnight following the manufacturing instructions. Fur instead of Sumo-Fur was used in competition experiment to better distinguish the shift bands between CcpE and Fur.

## Dye primer-based DNase I footprinting assay

Dye primer-based DNase I footprinting assay was performed as described previously [12,64]. Generally, DNA fragments were amplified using *S. aureus* Newman genomic DNA as a template. For amplification the promoter of *isdC*, a 359 bp fragment covering nucleotide -252 to

nucleotide +107 relative to the start codon of *isdC* was amplified with the primer *isdC*-foot-printing-F/*isdC*-EMSA-R. For the promoter of *citB*, a 697 bp fragment covering nucleotide -624 to nucleotide +73 relative to the start codon of *citB* was amplified with the primer *citB*-footprinting-F/*citB*-EMSA-R. For the promoter of *pycA*, a 369 bp fragment covering nucleotide -469 to nucleotide -101 relative to the start codon of *citB* was amplified with primers *pycA*-footprinting-F/*pycA*-footprinting-R. For the promoter of *pxpB*, a 265 bp fragment covering nucleotide -226 to nucleotide +39 relative to the start codon of *citB* was amplified with primers *pxpB*-footprinting-F/*pxpB*-footprinting-R. For the promoter of *citZ*, a 370 bp fragment covering nucleotide -359 to nucleotide +11 relative to the start codon of *citZ* was amplified with primers *citZ*-footprinting-F/*citZ*-footprinting-R. All the DNA fragments were purified with a Cycle-Pure Kit (Omega, Catalog no. D6492-02).

For foot-printing assay, 300 ng purified DNA fragment, indicated amount of proteins (6×His-CcpE or Sumo-Fur) and the corresponding binding buffer was added to a total volume of 50 μL. The composition of the binding buffer for Sumo-Fur is as follows: 10 mM Tris-borate (pH 8.0), 1 mM $MgCl_2$, 25 mM KCl, 100 μM $MnCl_2$, and 50 μg/mL BSA. For 6×His-CcpE, the binding buffer includes: 20 mM Tris-HCl (pH 7.4), 50 mM KCl, 20 mM $MgCl_2$, 1 mM EDTA, 5% NP-40, 5% glycerol, and 1 mM DTT. To study the effect of citrate on the protection region of 6×His-CcpE for the *pycA*, *pxpB* or *citZ* promoter, 20 mM citrate was added to the buffer as needed.

For the reaction, the mixtures were incubated for 25 min on ice (for 6×His-CcpE) or 20 min at room temperature (for Sumo-Fur), followed by adding of 0.01 Unit RQ1 RNase-Free DNase I (Promega, Catalog no. M610A). DNase I digestion was carried out at 30˚C for 3 min and quenched with 90 μL stopping buffer (200 mM NaCl; 30 mM EDTA; 1% SDS). The mixtures were further extracted with 200 μL phenol: chloroform: isoamylalcohol (v:v:v, 25:24:1), and aqueous phase containing DNA fragment was precipitated with equal volume of isopropanol, washed twice with ice-cold 70% ethanol (v/v), and dried under vacuum. Finally, 10 μL $ddH_2O$ was used to dissolve the digested DNA fragment. Next, 5 μL digested DNA was mixed with 4.9 μL HiDi formamide and 0.1 μL Genescan-500 LIZ size standards (Applied Biosystems). A 3730XL DNA analyzer was used to detect the sample, and the result was analyzed with Gene-Mapper software (Applied Biosystems). A dye primer based sequencing kit (Thermofisher, Catalog no. 79260) was applied to accurately determine the sequences of protection region, and the corresponding label-free promoter DNA fragment was used as a template in the sequencing reaction. Electropherograms were viewed with GeneMarker v1.91 (Applied Biosystems).

## Rapid amplification of sequences from the 5' ends of mRNAs (5'-RACE)

*S. aureus* Newman strain was grown to mid-log phase and harvested by centrifugation. Total RNA was isolated and the transcription start site identification of *pycA* was identified using the 5′-RACE kit (TAKARA). 5′-RACE experiment was performed according to the instructions of the manufacturer. Generally, reaction without reverse transcriptase was used as the negative control. The synthesis of first-strand cDNA initiated from gene-specific reverse primer *pycA*-race-1. Using terminal deoxynucleotide transferase, an oligo-dC tail was then added to the 3′-end of the cDNA. Subsequently, direct amplification of tailed cDNA using nested primer *pycA*-race-2 and an abridged anchor primer. PCR products were cloned into Stellar Competent Cells and sequenced using the universal primer M13F of the T vector. The first nucleotide following the oligo-dC sequence was taken as the transcriptional start.

## *In vitro* transcription assay

For *in vitro* transcription of *citB*, a 662 bp fragment covering nucleotide -318 to nucleotide +344 relative to the start codon of *citB* was amplified with the primers *citB*-trans-F/*citB*-trans-

R. For *in vitro* transcription of *pycA*, a 757 bp fragment covering nucleotide -469 to nucleotide +288 relative to the start codon of *pycA* was amplified with the primers *pycA*-trans-F/*pycA*-trans-R. For *in vitro* transcription of *isdC*, a 800 bp fragment covering nucleotide -480 to nucleotide +320 relative to the start codon of *isdC* was amplified with the primers *isdC*-trans-F/*isdC*-trans-R.

All the DNA fragments was amplified form *S. aureus* Newman genomic DNA, purified using a Cycle-Pure kit (Omega, Catalog no. D6492-02), dissolved in RNase-free ddH$_2$O, and stored at -20˚C until used as template for *in vitro* transcription assay. In general, RNA polymerase holoenzyme (RNAP) was reconstituted by incubating the *E. coli* RNAP$^{core}$ (1U/μl, NEB, Catalog no.M0550S) with same volume of *S. aureus* SigA (28 μM as stock) on ice for 15 min prior to use. For *in vitro* transcription, DNA template (10 nM) was pre-incubated with reconstituted RNAP (2 μL) in transcription buffer (final volume 20 μL) containing 40 mM Tris-HCl, pH 7.5; 150 mM KCl; 10 mM MgCl$_2$; 1 mM DTT; 0.01% Triton X-100; 0.1 mg/mL bovine serum albumin and 10 U RNasin (RNase inhibitor, Solarbio, Catalog no. R8060) for 10 min on ice. To evaluate the impact of 6×His-CcpE or Sumo-Fur on transcription, the indicated concentrations of 6×His-CcpE or Sumo-Fur was pre-incubated with the DNA for 20 min on ice before the addition of RNAP. When necessary, MnCl$_2$ or citrate was included in transcription buffer at a final concentration of 100 μM and 20 mM, respectively.

Transcription was initiated by the addition of 1 μL nucleotide mixtures (NEB, Catalog no. N0466S), and the reaction mixtures were incubated at 37˚C for 15 min. Subsequently, 1U RNase free DNase RQ1 (Promega, Catalog no. M6101) was used to digest the template DNA according to the manufacture's instruction. After which, transcripts were recovered by RNA Extraction Reagent (Solarbio, Catalog no. P1011), and precipitated with isopropanol. The precipitates were washed twice with ice cold 70% ethanol and dried by vacuum centrifugal concentration. Then, the dried transcripts were dissolved in 200 μL RNase free ddH$_2$O and reversely transcribed to cDNA with the PrimeScript RT reagent kit (Takara, Catalog no. RR037A) with random primers. Finally, cDNA was used as a template to quantitate the transcripts. For quantitation transcripts of *isdC*, *citB* and *pycA*, primer pairs of *isdC*-RT-F/*isdC*-RT-R, *citB*-RT-F/*citB*-RT-R and *pycA*-RT-trans-F/*pycA*-RT-trans-R, respectively, were used. Groups without RNAP were included as a negative control, and the results were normalized to groups with RNAP but no 6His-CcpE or Sumo-Fur.

### Preparation of anti-IsdC and Western blot analysis

The DNA fragment coding the 29–192 amino acid residues of IsdC was PCR amplified using primers pET28a-*sumo*::*isdC*$_{29-192}$-F/pET28a-*sumo*::*isdC*$_{29-192}$-R, and ligated to pET28a-Sumo to generate pET28a-*sumo*::*isdC*$_{29-192}$. The Proteins were expressed in *E. coli* Rosetta (DE3) after induction with 0.8 mM IPTG at 37˚C for 4 h and then were purified by a His Trap HP column. Purified Sumo-IsdC$_{29-192}$ was submitted to immune Japanese White Rabbit at day 1, 12, 26, 40 with 0.3 mg, 0.15 mg, 0.15 mg and 0.15 mg proteins, respectively. At day 52, serum of rabbit was used to enrich anti-IsdC antibody by affinity purification. Concentrated antibodies were obtained with a concentration of 2 mg/mL and were stored in 50% glycerol in PBS buffer. Anti-IsdC antibodies were prepared by ABclonal Biotechnology Co., Ltd.

To detect the protein abundance of IsdC, overnight cultures were sub-cultured with an initial OD$_{600}$ of 0.1 into 10 mL fresh TSB medium supplemented with 1 mM 2,2'-bipyridyl (DIP), with a tube volume to medium volume = 5:1. After 5 h growth at 37˚C with shaking at 250 rpm, cells were harvested and the pellets were washed once with 1×TE buffer (10 mM Tris, pH 8.5; 1 mM EDTA). The pellets were further suspended in 1×TE buffer to an OD$_{600}$ of 6.0, ten units lysostaphin (Sigma, Catalog no. 9011-93-2) were added to 100 μL suspensions and the

mixtures were incubated at 37˚C with gentle shaking for about 30 min to obtain cell lysate. 25 μL 5×SDS loading buffer (250 mM Tris-HCl, pH 6.8; 10% SDS; 0.5% bromophenol blue; 50% glycerol; and 100 mM DTT) was added to the 100 μL cell lysate and the mixtures were heated at 100˚C for 10 min. SDS-PAGE was performed with a 10% polyacrylamide gel at 150 V for about 60 min. Proteins on the gel was transferred to PVDF membrane (Bio-Rad, Catalog no. 1620177) by semi-dry transfer for 20 min at room temperature. Proteins on the membrane was blocked with 5% (w/v) skimmed milk (prepared in TTBS buffer:10 mM Tris-HCl, pH 7.5; 150 mM NaCl; 0.04% Tween 20) at room temperature for 2.5 h with gentle shaking and then incubated with the primary anti-IsdC with a 1:1000 dilution at 4˚C overnight. Following wash out the non-specific binding of primary antibody, a 1:5000 diluted anti-rabbit IgG conjugated to horseradish peroxidase (Cwbiotech, Catalog no. CW0103S) was added and incubated at room temperature for 2 h with gentle shaking. Chemiluminescence was detected by Tanon-5200 multi according to the manufacturing instructions and band intensity was analyzed by using Image J software.

## *S. aureus* growth in TSB with or without DIP

Overnight cultures of *S. aureus* Newman or its isogenic mutants was transferred into 10 mL fresh TSB medium supplemented with or without 1 mM DIP in a 50 mL tube. All the strains were adjusted with an initial $OD_{600}$ of 0.1 and sub-cultured at 37˚C with shaking at 250 rpm. Cultures were sampled at indicated time points, and the optical density at 600 nm ($OD_{600}$) was monitored using a NanoDrop 2000c (Thermo Scientific) spectrophotometer.

## Growth of *S. aureus* in chemical defined medium (CDM)

Chemical defined medium (CDM) was prepared as previously described [65], except that L-aspartic acid (Asp) was omitted. In general, 1 L of this medium contained the following nutrients: L-cystine 0.24 g, L-glutamic acid 2.4 g, L-proline 2.4 g, L-arginine 0.36 g, L-glycine 2.4 g, L-histidine 0.48 g, L-lysine HCl 0.6 g, L-serine 2.4 g, L-valine 0.48 g, L-tyrosine 0.18 g, L-threonine 2.4 g, L-alanine 2.4 g, L-isoleucine 0.6 g, L-leucine 0.6 g, L-phenylalanine 0.2 g, L-tryptophan 0.06 g, L-methionine 0.18 g, D-glucose 4.0 g, $(NH4)_2SO_4$ 6.84 g, nicotinic acid $11.94\times10^{-3}$ g, thiamine HCl $6.0\times10^{-4}$ g, $KH_2PO4$ 1.34 g, $NaHPO_4$ 5.7 g, $CaCl_2.2H_2O$ $5.0\times10^{-3}$ g, $CoCl_2.6H_2O$ $2.4\times10^{-3}$ g, $CuSO_4.5H_2O$ $5.0\times10^{-4}$ g, $MnSO_4.4H_2O$ $5.6\times10^{-3}$ g, NaCl $5.8\times10^{-3}$ g, $ZnCl_2$ $3.4\times10^{-3}$ g, $FeCl_3.6H_2O$ $2.7\times10^{-2}$ g, $MgSO_4. 7H_2O$ $2.47\times10^{-1}$ g. The resulting CDM was defined as CDM (-Asp). L-Aspartic acid (Asp, CAS no. 56-84-8) was dissolved with NaOH as a 100× stock solution, and adjusted with HCl to pH 7.5 for use. To evaluate the influence of Asp on the growth ability of strains, overnight cultures of *S. aureus* (in TSB) strains were washed once with CDM (-Asp) and transferred to fresh CDM supplemented without or with Asp (final concentration: 2.4 g/L) with an initial $OD_{600}$ of 0.05. 150 μL bacteria samples were triplicately added to 96 well plate with a transparent bottom, and 60 μL filter-sterilized mineral oil was added to prevent evaporation during the growth at 37˚C. The $A_{600}$ absorption was detected by a Synergy 2 Multi-mode Microplate Reader with an interval of 1 h.

## Intracellular citrate concentration determination

Liquid-state nuclear magnetic resonance (NMR) was used to detect the intracellular citrate concentration of bacteria. Generally, overnight cultures were transferred to 10 mL fresh TSB medium (tube volume to medium volume = 5:1) with an initial $OD_{600}$ of 0.1. After 6 h sub-cultivation at 37˚C with shaking at 250 rpm, cultures were centrifuged at 4˚C to discard the supernatant. Cell pellets were flash frozen in liquid nitrogen and stored in -80˚C until NMR analysis. To quantify the intracellular citrate concentration, bacteria cell pellets were washed

twice with ice cold PBS buffer (137 mM NaCl, 2.7 mM KCl, 8 mM $Na_2HPO_4$, and 2 mM $KH_2PO_4$), resuspended in 1 mL $D_2O$ dissolved PB buffer (410 mM $Na_2HPO_4$, 50 mM $NaH_2PO_4$, in 100 mL $D_2O$, pH = 7.4) and homogenized with a FastPrep-24 sample preparation system (MP Biomedicals, 6.5 M/s, 40 s one time, three times in total with 3 min interval on ice between each homogenization). Subsequently, the homogenized mixtures were centrifugated at 4°C, 16,000 g for 5 min to remove the cell debris. 800 μL supernatant was used to quantify the citrate concentration at an analytical platform based on a liquid high-resolution Bruker Avance-III NMR spectrometer equipped with a high-sensitivity cryogenic probe operating at a frequency of 600.17 MHz for 1H observation at 298 K. Different concentrations of citrate (Shanghai Urchem Limited, CAS no. 6132-04-3), including 5 μM, 10 μM, 20 μM, 40 μM, 80 μM, 160 μM, 320 μM, 640 μM, 1280 μM were prepared with $D_2O$, and used for the construction of standard curves. Intracellular citrate concentration was calculate based on the assumptions that *S. aureus* cell volume is $5 \times 10^{-13}$ mL [66] and one $OD_{600}$ corresponds to $2 \times 10^8$ cells/mL [67].

Citrate assay kit (Abnova, Catalog no. KA0864) was also used for the determination of intracellular citrate concentration. In this circumstance, overnight cultures were sub-cultured with an initial $OD_{600}$ of 0.1 into 10 mL fresh TSB medium supplemented with or without 1 mM DIP, and with a tube volume to medium volume = 5:1. All Cultures were sampled after 2 h cultivation at 37°C with shaking at 250 rpm and washed twice with ice cold PBS buffer, followed by resuspended in citrate assay buffer (provided in the assay kit) with an $OD_{600}$ of 10. Thirty Units lysostaphin (Sigma, Catalog no. 9011-93-2) were added to 300 μL suspensions, and the mixtures were incubated at 37°C with gentle shaking for 15 min to lyse bacterial cells. Clear lysate was submitted to citrate concentration quantification according to the manufacturing instructions.

## Aconitase activity assays

Overnight cultures of *S. aureus* Newman and $^{Tn}citB$ were transferred to 50 mL fresh TSB (flask volume to medium volume = 5:1) without or with 1 mM DIP (for Newman) with an initial $OD_{600}$ of 0.1. All of the sub-cultures were cultivated at 37°C with shaking at 250 rpm for 2 h. Cultures were centrifugated at 4°C to discard the supernatant. Cell pellets were washed twice with ice-cold PBS buffer and resuspended with 1 mL ice-cold PBS buffer. Suspensions were submitted to homogenization by a Bullet Blender Storm tissue homogenizer (Next Advance). Supernatant was collected after centrifugation at 4°C, 16,000 g for 5 min. Aconitase activities were assayed with an Aconitase Assay kit (Sigma, MAK337) according to the manufacturing instructions. Bicinchoninic acid (BCA) protein assay kit (meilunbio, MA0082) was used to quantify the protein concentration of the supernatant, and aconitase activities were normalized to total proteins of samples.

## Real-time quantitative PCR (RT-qPCT) analysis

Total RNA was extracted by a QIAGEN RNeasy extraction kit (Catalog no. 74104) following the manufacture's recommendations. Subsequently, total DNase-treated RNA (~400 ng) was reversely transcribed to cDNA with the PrimeScript RT reagent kit (Takara, Catalog no. RR037A) with random primers. For RT-qPCR analysis, 10 μL 2×Hieff qPCR SYBR Green Master Mix (Yeasen Biotech. Co.Ltd.), 300 nM primers and 2 μL diluted cDNA (as template) was used, $ddH_2O$ was added to a final volume of 20 μL. RT-qPCR was performed using a CFX Connect Real-Time PCR Detection System (Bio-Rad). Dissociation curve was analyzed to check the product homogeneity. The primer pairs used for RT-qPCR of *citB*, *pycA*, *pxpB*, *isdC*, *isdA*, *sirA*, *sbnA*, and 16S rRNA are *citB*-RT-F/*citB*-RT-R, *pycA*-RT-F/*pycA*-RT-R, *pxpB*-RT-F/

*pxpB*-RT-R, *isdC*-RT-F/*isdC*-RT-R, *isdA*-RT-F/*isdA*-RT-R, *sirA*-RT-F/*sirA*-RT-R, *sbnA*-RT-F/*sbnA*-RT-R and 16S-RT-F/16S-RT-R, respectively. Amplicon of 16S rRNA was used as the internal control. Expression levels of targeted genes were calculated by relative quantification method ($\Delta\Delta$CT) as previously described [12].

### β-galactosidase assays

Overnight cultures of the promoter-*lacZ* reporter strains were sub-cultured into fresh TSB medium with a final optical density 600 nm ($OD_{600}$) = 0.1. Liquid cultures were grown in a 50-mL tube with a volume to medium ration of 5:1, shaking at 37˚C with 250 rpm. Cultures were sampled at the indicated time points and bacteria pellets were washed once with Z-buffer (60 mM $Na_3PO_4.7H_2O$, 40 mM $NaH_2PO_4.H_2O$, 10 mM KCl, 1 mM $MgSO_4.7H_2O$, pH7.0). Subsequently, β-galactosidase activity assays were performed according to the method described previously [12]. Product (7-hydroxy-4-methylcoumarin, 4 MU) was detected with an excitation/emission wavelength of 360 ± 20 nm/460 ± 20 nm respectively, using a Synergy 2 (Biotek) plate reader following the manufacturer's instructions. Samples were detected in triplicate and relative LacZ activity was normalized based on $OD_{600}$.

### Mouse infection models

Overnight cultures of *S. aureus* strains were diluted 100-fold in 50 mL fresh TSB medium in a 250 mL flask. Sub-cultures were performed at 37˚C with shaking at 250 rpm for 3 h. Bacteria pellets were collected by centrifugation, washed twice and resuspended with sterile ice-cold phosphate-buffered saline (PBS). Colony formation units (CFUs) were enumerated before mice were infected.

Mouse infections were performed as described previously [12,68,69]. Briefly, 6 to 8-week-old female BALB/c mice purchased from Shanghai SLAC Laboratory Animal Co. Ltd. were housed under specified pathogen-free conditions. Before infection, mice were anaesthetized with pentobarbital sodium (80 mg/kg) by intraperitoneal injection. Anaesthetized mice were infected retro-orbitally with *S. aureus*. Four days post-infection, the animals were euthanized and, their kidneys and hearts were aseptically removed and homogenized in 1 mL sterile PBS. Serial dilutions of each organ were then plated on TSA plates to enumerate the colony-forming units (CFUs). Statistical significance was determined using the two-tailed Mann-Whitney test.

### Statistical analysis

Statistical analysis was conducted using GraphPad Prism version 7.00 software. Two-tailed one-sample t-test, Student's two-tailed unpaired t-test, One-way ANOVA, Two-way ANOVA or Mann-Whitney two-tailed test were used to compare datasets where appropriate.

## Supporting information

**S1 Fig. Intracellular citrate levels of *S. aureus* strains.** The intracellular citrate levels of *S. aureus* Newman and its isogenic mutants cultured either in TSB for 6 h (A) or in TSB with 1 mM 2,2-dipyridyl (DIP) for 2 h (B), measured by citrate assay kit. Newman, $^{Tn}citB$, $\Delta citZ^{Tn}citB$ and $\Delta sbnG\Delta citZ^{Tn}citB$ harbor an empty control vector, respectively. $\Delta sbnG\Delta citZ^{Tn}citB$/p-*sbnG* represents the $\Delta sbnG\Delta citZ^{Tn}citB$ strain complemented with *sbnG*. Data represents mean ± SD from *n* = 3 biological replicates, *n.s.*, *p* > 0.05; *** *p* < 0.001 by One-way ANOVA with Tukey test. *n.d.*, not detectable.
(TIF)

**S2 Fig. OPLS-DA analysis of electrospray ionization features and volcano plots for differentially expressed genes.** (A and B) Orthogonal partial least squares-discriminant analysis (OPLS-DA) of electrospray ionization (ESI) features generated from ultra-performance liquid chromatography-coupled-quadrupole time of flight mass spectrometry (UPLC-Q-TOF-MS/MS) in positive (A) and negative (B) ionization mode. Data represents $n$ = 6 biological replicates for Newman, $^{Tn}citB$, $\Delta citZ^{Tn}citB$ and $\Delta ccpE^{Tn}citB$, as well as $n$ = 3 for the pooled quality control (QC) samples. (C) (DEGs) between $\Delta citZ^{Tn}citB$ and $^{Tn}citB$ mutants. X-axis and y-axis represent $\log_2$ fold-change differences and statistical significance (present as the negative log of DEG $p$-values), respectively. Significantly up-regulated and down-regulated genes are indicated by red and blue dots, while non-significant genes are shown as grey dots.
(TIF)

**S3 Fig. Results of joint-pathway analysis.** (A) Identification of the citrate-regulated pathway through transcriptomics-metabolomics coupled analysis. (B) Identification of CcpE-regulated pathway through transcriptomics-metabolomics coupled analysis. The x axis represents pathway impact scores, which summarize normalized topology measures of the perturbed genes/metabolites within each pathway. The y axis displays the $-\log_{10}(P)$ values derived from the enrichment analysis results. The size of each data points is proportional to its x-axis values, and the color gradients correspond to their y values. The dashed line signifies a significance threshold, corresponding to a $p$ value equal to 0.05, which equates to a $-\log_{10}(P)$ of 1.30.
(TIF)

**S4 Fig. A schematic presentation of the top five IDAP-Seq peaks for CcpE.** Images represent the representative of CcpE IDAP-seq data showing the binding of CcpE to the promoter region of *pycA* (A), the promoter region of *pxpB* (B), the promoter region of *citZ* (C), the *NWMN_2386* open reading frame region (D), and the promoter region of *citB* (E). IDAP-seq data originated from replicate 1 (Biosample accession number: SAMN24812582) was used to generate the images with Integrated Genome Viewer (IGV) software. The fold enrichment and $-\log_{10}^{(q\text{-value})}$ of the IDAP-seq peaks produced by MACS2 were presented. The ORF arrows below the IGV images indicate the direction of transcription (left or right). The gene names (or locus tag) are indicated above the ORF arrows, while the relative expression (fold-change) is indicated below the ORF ($^*p < 0.05$ by edgeR exact test). Potential CcpE-targeted genes are highlighted by yellow arrows. The IVG visualization of RNA-seq data for *citB* loci is also presented in (E), and the transposon insertion site of the $^{Tn}citB$ mutant is marked by a red arrow.
(TIF)

**S5 Fig. Schematic overview of the regulatory influence of citrate accumulation and CcpE on the metabolism of *S. aureus*.** The pink shading indicates glycolysis/gluconeogenesis, the light-yellow shading indicates the pentose phosphate pathway, the blue shading indicates the TCA cycle, and the green shading indicates the phosphoenolpyruvate-pyruvate-oxaloacetate (PPO) node. The arrows, from left to right, respectively indicated $\Delta citZ^{Tn}citB/^{Tn}citB$ and $\Delta ccpE^{Tn}citB/^{Tn}citB$. The upwards arrow in red, denotes upregulation, and the downwards arrow in blue, denotes downregulation. The 12 well-known branchpoint metabolites that serve as exit points for carbon from the central carbon metabolism network are underlined [32]. The direct reactions are indicated by continuous line with an arrow at the end, while multiple steps of reaction are indicated by dashed lines with an arrow at the end. The genes of interest are notated in italics. The asterisks (*, **, and ***) indicates a gene or operon whose promoter region was bound by CcpE in one, two, and three independent IDAP-seq experiments, respectively. PEP, phosphoenolpyruvate; CoA, coenzyme A; PRPP, 5-Phosphoribosyl diphosphate; Fru-1,6-2P, fructose 1,6-bisphosphate; DHAP, dihydroxyacetone; GlcN-6P, Glucosamine

6-phosphate; GlcN-1P, Glucosamine 1-phosphate; GlcNAc, N-Acetyl-glucosamine; GlcNAc-6P, N-Acetyl-glucosamine 6-phosphate; UDP-GlcNAc, UDP-N-Acetyl-glucosamine; UDP-MurNAc, UDP-N-Acetylmuramic acid; Man-6P, Mannose 6-phosphate; ManNAc, N-Acetyl-mannosamine; ManNAc-6P, N-Acetyl-mannosamine 6-phosphate; Neu5Ac, N-Acetylneura-minate.
(TIF)

**S6 Fig. CcpE binds to the promoter of Fur-regulated genes in *S. aureus*.** Representative images of CcpE IDAP-seq data showing the binding of CcpE to the promoter region of *isdB* (in A), the intergenic region between *isdA* and *isdC* (in A), the promoter region of *isdI* (in B), the promoter region of *fhuABG* operon and *fhuD2* (in C), and the intergenic region between *sirA* and *sbnA* (in D). IDAP-seq data originated from replicate 1 (Biosample accession number: SAMN24812582) was used to generate the images with Integrated Genome Viewer (IGV) software. The fold enrichment and $-\log_{10}^{(\text{q-value})}$ of the IDAP-seq peaks generated from MACS2 were presented. Arrows below the IGV images represent ORFs pointing left or right, indicating the direction of transcription. Gene names (or locus tag) are shown above the ORF arrows. Numbers below the ORF are relative expression levels (fold-change) ($^*p < 0.05$ by edgeR exact test). The IVG visualization of RNA-seq data for the *isd* loci, as depicted in (A), reveals the presence of a potential internal promoter situated upstream of the *isdF-srtB-isdG*. This internal promoter may account for the observed differential expression pattern of the *isd* operon genes.
(TIF)

**S7 Fig. Experiments showing that CcpE modulates the Isd system, and citrate accumulation activates the regulatory function of CcpE.** (A) EMSA result shows that Fur can compete CcpE for binding to the *isdC* promoter region. (B) Quantitative analysis of Western blotting for IsdC in Fig 4E. Glyceraldehyde 3-phosphate dehydrogenase (GAPDH) was used as the loading control. The results are reported as relative abundance with the control sample (Newman) set to 1. (C) Growth curve of *S. aureus* Newman in TSB without or with 1 mM DIP. (D) Intracellular citrate levels of *S. aureus* strains cultured in TSB without or with DIP for 2 h. (E) Aconitase activities of WT Newman and $^{\text{Tn}}$*citB* mutant. (F) RT-qPCR analysis of *citB* transcripts in WT *S. aureus* Newman strain grown in TSB without or with 1 mM DIP. Results were reported as fold changes with the 2 h (without DIP) group set to 1. In (B), data represents mean ± SD from $n = 5$ biological replicates. In (C to F), data represents mean ± SD from $n = 3$ replicates. $^*p < 0.05$, $^{**}p < 0.01$, $^{***}p < 0.001$, by two-tailed one-sample *t*-test when compared with the Newman control where set to 1 (B and F), otherwise two-tailed unpaired *t*-test was used.
(TIF)

**S8 Fig. Fur activates the expression of *citB in vitro* and *in vivo*.** (A) EMSA shows that Fur binds to *citB* promoter *in vitro* in a concentration dependent manner. The *citB*-p fragment contains a DNA sequence covering nucleotide -624 to nucleotide +73 relative to the start codon of *citB*. An DNA fragment containing the coding region of *murA* gene was served as control. (B) The protection pattern of the *citB* promoter after digestion with DNase I following incubation in the absence and presence of Fur *in vitro*. The protected regions (relative to the start codon) are underlined. (C) Fur enhances *citB* mRNA expression *in vitro*. The transcripts were extracted and quantified by RT-qPCR. Data represents mean ± SD from $n = 3$ independent experiments. Statistical analysis was performed using two-tailed one-sample *t*-test compared with the RNAP group, which was set to 100% ($^*p < 0.05$, $^{***}p < 0.001$). (D and E) *citB*-*lacZ* activity in *S. aureus* strains grown in TSB for 4.5 h. The relative fluorescence intensity (RFU) was normalized to the optical density of bacteria at 600 nm ($OD_{600}$). Data represents mean ± SD from $n = 3$ biological replicates and statistical analysis was performed using

Student's two-tailed unpaired *t*-test (***$p < 0.001$).
(TIF)

**S1 Table. Results of Metabolome.**
(XLSX)

**S2 Table. Results of RNA-seq.**
(XLS)

**S3 Table. Results of IDAP-seq.**
(XLS)

**S4 Table. Sequences used for MEME analysis of the binding motif for CcpE.**
(DOCX)

**S5 Table. Plasmids and strains used in this study.**
(DOCX)

**S6 Table. Primers used in this study.**
(DOCX)

## Author Contributions

**Conceptualization:** Feifei Chen, Lefu Lan.

**Data curation:** Feifei Chen, Qingmin Zhao, Ziqiong Yang, Xia Liu.

**Formal analysis:** Feifei Chen, Qingmin Zhao, Ziqiong Yang, Jianhua Gan, Naixia Zhang, Cai-Guang Yang, Haihua Liang, Lefu Lan.

**Funding acquisition:** Feifei Chen, Lefu Lan.

**Investigation:** Feifei Chen, Qingmin Zhao, Ziqiong Yang, Rongrong Chen, Huiwen Pan, Yanhui Wang, Huan Liu, Qiao Cao, Xia Liu.

**Methodology:** Feifei Chen, Jianhua Gan, Haihua Liang, Lefu Lan.

**Resources:** Feifei Chen, Naixia Zhang, Cai-Guang Yang, Haihua Liang, Lefu Lan.

**Supervision:** Lefu Lan.

**Writing – original draft:** Feifei Chen, Lefu Lan.

**Writing – review & editing:** Feifei Chen, Lefu Lan.

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
