## [Decision Letter · Decision Letter 0]

21 Nov 2023

Dear Dr Chen,

Thank you very much for submitting your manuscript "Citrate serves as a signal molecule to modulate carbon metabolism and iron homeostasis in Staphylococcus aureus" for consideration at PLOS Pathogens. As with all papers reviewed by the journal, your manuscript was reviewed by members of the editorial board and by several independent reviewers. The reviewers appreciated the importance of your study but they noted sme issues to solve. Two reviewers noted that many data were only gnerated using TCA mutants and they request additional experiments with wild-type strains. All reviewers suggesting additional controls and better presentation and discussion of the results. Please imporve the flow and narrative of the mansucript, which appears to be blurred by the large amoount of presentated details. In light of the reviews (below this email), we would like to invite the resubmission of a significantly-revised version that takes into account the reviewers' comments.

We cannot make any decision about publication until we have seen the revised manuscript and your response to the reviewers' comments. Your revised manuscript is also likely to be sent to reviewers for further evaluation.

Sincerely,

Andreas Peschel, Ph.D.

Academic Editor

PLOS Pathogens

Michael Otto

Section Editor

PLOS Pathogens

Kasturi Haldar

Editor-in-Chief

PLOS Pathogens

orcid.org/0000-0001-5065-158X

Michael Malim

Editor-in-Chief

PLOS Pathogens

orcid.org/0000-0002-7699-2064

Reviewer's Responses to Questions

**Part I - Summary**

Reviewer #1: In the manuscript “Citrate serves as a signal molecule to modulate carbon metabolism and iron homeostasis in Staphylococcus aureus”, Chen et al. extend a previous study on the CcpE regulatory protein that was shown to bind citrate and allosterically regulate its DNA-binding activity, resulting in the differential expression of dozens of genes. CcpE is a major positive regulator of TCA-cycle activity, but also regulates the expression of staphylococcal virulence genes. A growing body of evidence demonstrates the linked nature of physiology and virulence; thus, a better understanding of the molecular mechanisms underpinning these linkages is significant and may lead to new therapies to combat this potentially devastating human pathogen.

In the current study, the authors use a citB mutant (blocked for the TCA cycle and accumulates citrate) and a citB citZ mutant (blocked for the TCA cycle, does not accumulate citrate) to interrogate the transcriptome, metabolome, and virulence phenotypes. This is a clever approach, as the comparison avoids confounding effects of disrupting the TCA cycle generally on cell physiology. In doing so, they identify differentially regulated genes dependent and independent of citrate levels, and metabolite changes dependent and independent of ccpE. This is definitely a resource for others in the field. They solve a crystal structure of CcpE’s regulatory domain with citrate, and identify additional amino acid residues that indirectly affect the binding of citrate. They also perform IDAP-Seq, a genome-wide DNA-binding assay to identify direct targets of CcpE. The authors identify a CcpE DNA-binding motif that is strikingly similar to the Fur binding motif. Admittedly the motif is highly degenerate (TWWDWWWWRVTTATCATT). They identify and they identify a subset of genes that are regulated by both Fur and CcpE, and use EMSA, footprinting, and in vitro transcription in an attempt to demonstrate direct binding and regulation. There is limited testing of the role of bases in the motif in transcription factor binding. In the end, the authors come up with a model whereby CcpE modulates iron homeostasis, connecting CcpE and Fur activities.

In the end, the manuscript is very long and dense, and it is very difficult to assimilate all the information. It seems as though there could be multiple, independent manuscripts here. Importantly, I was puzzled why the authors did not bring up the previously described second citrate synthase SbnG (https://pubmed.ncbi.nlm.nih.gov/24666349/). This seems like an important consideration, or at least one to mitigate early in the manuscript text if not important. Activity of a second citrate synthase might affect intracellular pools, at least under iron limitation. The structural studies included here seem incremental, and it is unclear what they really add to our knowledge of CcpE function, and the overall story herein. Generally, there are so many things being examined in the manuscript that the focus is lost, and the reader is left trying to understand the overall model. For instance, do CcpE and Fur compete for binding of target genes? Which motif(s) are required for these regulations? In many cases, there are missed opportunities to test hypotheses directly; instead, the authors stick to the same series of mutant strains and complement strains, and show correlations. In some places the data may be over-interpreted. At best, the results of those experiments suggest, not indicate, molecular mechanism. Overall, the writing could be made clearer, and there are a number of misspellings. The authors frequently use the phrase “To further investigate X, we did Y” – this is vague, and it would be more helpful to use more intentional language and clear about what they are testing. Finally, there is frequent use of t-tests to test statistical significance where one-way ANOVA and post test should be used, to avoid errors with multiple hypothesis testing.

Reviewer #2: In their manuscript, Chen et al. investigate the influence of citrate on the regulation of carbon and iron metabolism in S. aureus through CcpE, a LysR-type transcriptional regulator. They distinguish between direct and indirect CcpE targets using metabolomics, transcriptomics, and IDAP sequencing and determine the crystal structure of CcpE bound to citrate. The manuscript also discusses CcpE's negative regulation of pycA and its positive regulation of Fur-regulated genes. Citrate's role in iron homeostasis and its impact on Fur regulatory activity are emphasized. Additionally, the authors demonstrate citrate's effect on S. aureus pathogenesis through mouse and G. mellonella infection models. Overall, the study successfully highlights a very underappreciated role for citrate in modulating gene regulation in S. aureus.

The manuscript is extensive and dense, which can sometimes lead to confusion. In my opinion, the authors should consider condensing it.

Reviewer #3: Chen, F. and colleagues present a manuscript on the role of citrate as a regulatory molecule modulating carbon metabolism and iron homeostasis.

The authors show that mutant strains of S. aureus with altered intracellular levels of citrate (TCA cycle mutants) have strongly altered metabolic and transcriptional profiles. They show that citrate binds directly to the transcriptional regulator CcpE with diverse regulatory effects. Additionally, it is suggested that the ability of citrate to chelate metals (especially Fe and Mg) alters Fur-activity thereby impacting transcription of iron-acquisitions genes. Furthermore, CcpE and Fur are shown to have overlapping DNA recognition sites, further supporting the role of CcpE (and with it citrate) in modulating iron starvation responses. Finally, the effects of mutants are validated using animal models of infection.

The concept of this study adds a novel layer of complexity of iron starvation responses in S. aureus und is relevant for the readership of this.

This manuscript contains an enormous amount of data and experiments addressing the topic at with various technical approaches and all relevant physiological levels. The authors have to be congratulated to this comprising effort.

It has to be said, that some of the individual experiments are lacking individual controls that would strengthen the drawn conclusions. E.g. Complex formation of CcpE in the presence of citrate (Fig. 2a) using formaldehyde crosslinking is little convincing and appropriate ΔCcpE controls are not available. However, the authors show also crystals of citrate-CcpE complementing the deficit of the other experiment. Similarly, the authors show that Fur does not bind to its DNA target when excess citrate is available (Fig. 5e) claiming it to be caused by metal-chelation by citrate. Appropriate controls including equimolar citrate:metal concentrations are not included. However, in the general context of data showing intracellular metal levels, protein expression levels, transcriptional changes etc. these small flaws do not question the general drawn conclusions.

**Part II – Major Issues: Key Experiments Required for Acceptance**

Reviewer #1: Major issues:

1. Some sections are complex and difficult to parse. For instance: Lines 120-125. Please review the entire manuscript for other instances.

2. For differential changes in gene expression and metabolite levels, the phrasing citZ citB vs. citB is vague. Please use citZ citB / citB etc for comparisons to clarify the over- or under-expression ratios.

3. Fig 2 and Fig S4: Western blot resolution is poor and not convincing for demonstrating citrate alters oligomerization of CcpE.

4. Lines 233-235: The thermal shifts are rather small. Is this typical? Perhaps cite other examples where true binding affects thermal shift in this small amount.

5. Line 238: D256A variant – control western seems missing that shows that it is made to WT levels.

6. Lines 324-340: The data for CcpE controlling pycA seems confusing to me. Perhaps it’s the writing, but I don’t understand the model: the ccpE mutant has increased pycA transcript and pycA-lacZ activity, and CcpE addition to in vitro transcription assays blocks transcription. This is all consistent with CcpE as a repressor. However, mutating the T-N11-A motif results in a loss of the upregulation in both WT and ccpE cells.

7. Lines 353-355: this is a nice result, but somewhat indirect. Can you make a stronger conclusion by doing a more direct experiment? For instance, can you uncouple the expression of pycA from its regulation? Maybe express from an inducible or constitutive promoter?

8. CcpE modulates Fur-regulated genes section: On some level this becomes very descriptive, and dense. “To investigate further…” Why? What is a detail and what is important here? This reads as unfocused. Maybe clarify your model here, and design experiments to test specific hypotheses? Why the dual regulation by CcpE and Fur? Do they both use the same binding sites? Or one each? Mutagenesis can help here. Also: “To clarify the influence….” Is there a reason to expect post-transcriptional regulation? Again, isn’t the interesting and big question the mechanism how CcpE and Fur use the same site?

9. Lines 418-421: The authors use the word “suggest” here, but still seems overreaching. CcpE’s effect is on transcription, as far as we know. Anything else is speculative.

10. The Citrate accumulation and activation of CcpE section seems like it would be better placed before CcpE modulates iron homeostasis in citB mutant section. I was lost in the latter section until I came to the former section.

11. Animal experiments in Fig 6: there is significant dispersion in the data for some experiments; thus, it’s hard to make firm conclusions.

12. What is the purpose of the experiment in 6D? Seems peripheral to the story.

13. For as long and expansive the manuscript is, the discussion is relatively shallow and doesn’t really contextualize the data in a meaningful way.

Reviewer #2: 1. I understand the rationale behind utilizing the citB and citZcitB mutants throughout the manuscript. Nonetheless, it's important to note that both of these mutants exhibit partial impairment of the TCA cycle (as evidenced by post-experimental growth defects in Fig S1B), which introduces complexity into the interpretation of whether observed effects are solely attributable to citrate or influenced by impaired TCA cycle function or growth-related factors.

2. Regarding the citB and citZcitB mutants, it is theoretically expected that they should not be capable of producing physiological levels of citrate within cells. In the citB mutant, citrate levels are notably elevated (~16 mM), while in the citZcitB mutant, inactivation of citZ should lead to a complete depletion of the citrate pool, despite the authors reporting citrate levels close to those of the wild type.

a. It would be beneficial if the authors could confirm the intracellular source of citrate in the citZcitB mutant.

b. Given that these mutants may not accurately reflect physiological citrate levels, it would be valuable to provide evidence demonstrating that, under physiological concentrations in the wild-type, citrate can indeed function as a signaling molecule to regulate this extensive regulon.

3. Although the authors have used omics-based studies, there isn’t a good summarization of the major impact of CcpE on cellular metabolism. For instance, what are the major genes of central and peripheral metabolism that are directly controlled by CcpE? Do the differences in metabolism as found from metabolomics and RNA seq correlate with the direct targets of CcpE? The closest the authors have come to is give a summarization in Fig S6. However, this figure is not sufficiently discussed to give the readers what CcpE does globally in the cell.

4. Fig 2A and 2B- The tetramer to monomer ratio determination from immuno blots is not very convincing at all. First, there are no loading controls. Second, the dimer and tetramer bands are barely visible, making me wonder how robust the quantification in 2B is.

5. The finding that CcpE inhibits pycA is intriguing and Fig S7 s

---

## [Decision Letter · Decision Letter 1]

28 May 2024

Dear Dr Chen,

Thank you very much for submitting your manuscript "Citrate serves as a signal molecule to modulate carbon metabolism and iron homeostasis in Staphylococcus aureus" for consideration at PLOS Pathogens. As with all papers reviewed by the journal, your manuscript was reviewed by members of the editorial board and by several independent reviewers. The reviewers appreciated the attention to an important topic but Reviewers 1 and 2 indicated that the statistical analysis, interpretation, and presentation of data need to be improved. Please also address the concerns on clarity and readability of the text. Based on the reviews, we are likely to accept this manuscript for publication, providing that you modify the manuscript according to the review recommendations.

Sincerely,

Andreas Peschel, Ph.D.

Academic Editor

PLOS Pathogens

Michael Otto

Section Editor

PLOS Pathogens

Michael Malim

Editor-in-Chief

PLOS Pathogens

orcid.org/0000-0002-7699-2064

Reviewer Comments (if any, and for reference):

Reviewer's Responses to Questions

**Part I - Summary**

Reviewer #1: In the resubmission, the authors have addressed some, but not all of my concerns. Notably, the manuscript remains very dense, and while the study is definitely in my area of expertise, I had a very difficult time getting through the experiments and understanding the model and conclusions. I don't think the crystallography adds much to the story, and is incremental. The regulatory model with CcpE and Fur is not clear.

Reviewer #2: I want to thank the authors for addressing my initial concerns. The study is very valuable, novel and effectively showcases the impact of citrate on iron homeostasis and the metabolism of S. aureus. Upon reading the revised manuscript, I understand that the model presented indicates an increase in the intracellular pool of citrate has two major consequences for S. aureus. First, this activates CcpE through direct binding of citrate, leading to differential regulation of metabolism, particularly impacting certain nodes of central metabolism. Convincing evidence by the authors shows that citrate-activated CcpE competes with the Fur-binding site in promoter regions, leading to the activation of Fur-repressed genes. Separate from CcpE, the authors also demonstrate that citrate itself can decrease the repressor activity of Fur by depleting cells of divalent ions required for Fur function. Although the evidence for the latter consequence is strong, there are some aspects in this line of inquiry that could be further clarified.

Reviewer #3: The authors present an extensively modified manuscript. My comments were addessed sufficiently.

**Part II – Major Issues: Key Experiments Required for Acceptance**

Reviewer #1: • Lines 286-291: The model is difficult to conceptualize. CcpE acts as an apparent repressor, and knocking out ccpE results in de-repression. But this regulation is far from the core promoter and transcription +1. How does it work? Also, mutating doesn’t seem to tell you much about regulation by CcpE if the basal promoter activity is affected. Again, still unclear how this would work if it’s so far upstream of the promoter.

• Lines 318-320: If CcpE binds to promoters and enhances expression of isdA, etc., then I would expect increased expression in citB mutant, and less in citB citZ mutant, and less in citB ccpE mutant. Doesn’t appear to be the case with the values below the genes. (Fig S7)

• For CcpE binding to the isd promoters – looks like the binding is intrinsic and not dependent on citrate.

• Fig 5D – The figure legend lacks detail to help the reader to interpret the EMSAs. In Fig S8, what is meant by P2 form of IsdC?

• Lines 333-345: Fig 5E – the western blot isn’t convincing for showing the effect of citrate activating CcpE and activating isdC expression

• Lines 355-357: I don’t see evidence of this. CcpE appears to bind multiple Fur-regulated genes. RNA-Seq shows a positive effect of CcpE on isd gene expression

• Lines 362-373: With the DIP treatments, could it be that Fur-mediated repression is lifted? Wouldn’t you have to knock out ccpE to show that CcpE is activating under these conditions? I don’t think it’s clear that the authors have iron deficiency augments the regulatory function of CcpE in S. aureus

• Fig 6A and Lines 375-385: I’m confused here. If CcpE negatively regulates iron uptake and Fur negatively regulates uptake, then one would expect either no change or an increase in iron levels in the ccpE mutant, right? Why the change in the citB background? The changes are potentially statistically significant but are they biologically significant? Also, using Dunnett;s test compares samples to a control. This looks like a Tukey’s test. Please verify. I guess having a model would be helpful here.

• Fig 6F is not very convincing – authors state on Lines 411-412 “…citrate has the ability to alleviate the inhibition of isdC transcription by the recombinant Fur protein.” The effect is small.

• At the end of section, line 418: I’m lost…the manuscript is dense and the writing is not clear.

Reviewer #2: 1. While the authors have shown that CcpE differentially regulates various metabolic genes (Fig S6), it would be helpful to compare this to the metabolomics data provided and highlight its actual metabolic consequences to the cell. Also, a pathway enrichment analysis might also highlight some primary impacts of CcpE in S. aureus.

2. Line 112: The authors indicate that the intracellular pool of iron in the citB mutant was steady despite the citZ mutation and this surprising finding could not be attributed to sbnG. To me, this is confusing since there shouldn’t be any alternate source of citrate, or at least none the authors have identified. I wonder if what the authors are detecting by mass spectrometry is isocitrate in the cell, formed through the glutamate/2-oxoglutarate node within the TCA cycle. Note that isocitrate is not readily distinguished from citrate by mass spectrometry. If the authors have not ruled out isocitrate, they should state this possibility in the manuscript and reassess how this impacts the rest of their story. For instance, does isocitrate impact CcpE or chelate iron?

3. The most difficult aspect to interpret in the manuscript is the impact that the citZ mutation has on intracellular iron. If the authors’ interpretation that increased intracellular citrate in the citB mutant decreases the intracellular iron pool to impact Fur activity is true (Fig 6A), why does this mutant have the most sensitivity to Streptonigrin compared to the wild type? Even more confusing is the observation that the citZcitB double mutant is significantly less sensitive than WT to Streptonigrin, suggesting that iron levels are very low in the citZcitB mutant despite less citrate (Fig 6B). The authors need to measure the levels of intracellular iron in the WT and citZ mutants if readers are supposed to make sense of the Streptonigrin experiments.

Reviewer #3: (No Response)

**Part III – Minor Issues: Editorial and Data Presentation Modifications**

Reviewer #1: • Line 70: Use of a review on Salmonella is odd, since there are several now that focus on S. aureus. Strongly suggest citing Richardson AR 2019 (DOI: 10.1128/microbiolspec.GPP3-0011-2018)

• Line 83: please rephrase: “senses and responds to citrate, an intermediate of the TCA cycle…”

• Line 104: elevating: change to elevates

• Fig S1: It would help to have the y-axes at the same scale to make comparisons easier.

• Fig 1D: I still find the Venn diagrams difficult to interpret. Perhaps a better figure legend describing the comparisons and what is meant by citrate-regulated and CcpE-regulated would help…or setting up the comparisons in the manuscript body text

• Line 159: a round – change to around or better, approximately

• Lines 179-218: Still don’t know the structure-function studies add anything – seems more of a detail. And, it seems to interrupt the flow – gene expression, then structure, then binding site identification for direct targets.

• Line 234 and Fig S5A…check fold-enrichment values – they don’t match

• Fig 4AB, no citrate added? Label as such, why not? Stronger effect but did not test?

• Lines 295-314: These experiments are designed to test the hypothesis that repression by CcpE (when activated by high intracellular citrate) will generate phenotypic auxotrophy. The data are consistent, but the writing could be refined to make this clearer.

• Lines 311-314: Not clear on the logic: what the DIP treatment is doing here. Does it somehow increase citrate levels, leading to increased repression in a CitZ, SbnG, and CcpE-dependent manner? Please clarify.

• Lines 324-326: in vitro transcription Fig 5B – is this citrate dependent CcpE enhancement? Also stats should be ANOVA with Dunnetts test

• Fig S8B legend: is this the quantitative analysis of the western in Fig 5E? Please clarify in legend

• Lines 346-351: They don’t seem to contribute to the story and just make things less clear.

• Lines 371-373: Fur bound to iron typically acts as a repressor, but your data show a stimulatory role. This seems consistent with a report in E. coli for AcnA (aconitase). Please mention this. https://www.ncbi.nlm.nih.gov/pmc/articles/PMC4167408/

• Regarding the above point, please add some background information on Fur and the mechanism of regulation.

• Line 437-438: Instead of “…independent of citB” do you mean in the citB mutant?

Reviewer #2: 1. Line 126: Change ‘incubation’ to ‘growth of S. aureus strains’.

2. Line 128: Describe the strains used here.

3. Line 288: Remove the word ‘was’.

4. Line 522-524: When oxygen is present, S. aureus does not thought to actively metabolize glucose through the TCA cycle due to CcpA repression. Thus, acetate is the primary byproduct of glucose metabolism.

5. Line 482: Correct ‘post-experimental’ to ‘post-transcriptional’.

Reviewer #3: (No Response)

PLOS authors have the option to publish the peer review history of their article (what does this mean?). If published, this will include your full peer review and any attached files.

Reviewer #1: No

Reviewer #2: No

Reviewer #3: No

Figure Files:

Data Requirements:

Reproducibility:

References:

---

## [Editor Report · Decision Letter 2]

15 Jul 2024

Dear Dr Chen,

We are pleased to inform you that your manuscript 'Citrate serves as a signal molecule to modulate carbon metabolism and iron homeostasis in Staphylococcus aureus' has been provisionally accepted for publication in PLOS Pathogens.

Best regards,

Andreas Peschel, Ph.D.

Academic Editor

PLOS Pathogens

Michael Otto

Section Editor

PLOS Pathogens

Michael Malim

Editor-in-Chief

PLOS Pathogens

orcid.org/0000-0002-7699-2064
---

## [Editor Report · Acceptance letter]

25 Jul 2024

Dear Dr Chen,

We are delighted to inform you that your manuscript, " Citrate serves as a signal molecule to modulate carbon metabolism and iron homeostasis in Staphylococcus aureus ," has been formally accepted for publication in PLOS Pathogens.

Best regards,

Michael Malim

Editor-in-Chief

PLOS Pathogens

orcid.org/0000-0002-7699-2064